# Neural circuit models for evidence accumulation through choice-selective sequences

Lindsey S. Brown [1] ✉, Jounhong Ryan Cho[1], Scott S. Bolkan[1], Edward H. Nieh[1,5], Manuel Schottdorf [1,6], David W. Tank [1], Carlos D. Brody [1,2], Ilana B. Witten [1,2] ✉ & Mark S. Goldman [3,4] ✉

Decision making is traditionally thought to be mediated by neurons that accumulate evidence through persistent activity. However, recent decision-making experiments in rodents have observed neurons across the brain that fire sequentially, rather than persistently, with the subset of neurons in the sequence depending on the animal's choice. We developed two candidate circuit models in which neurons are active sequentially and transfer evidence faithfully to the next active population. One model encodes evidence in the relative firing of two competing chains of neurons, and the other in the network location of a stereotyped, bump-like pattern of neural activity. Neural recordings from four brain regions during an evidence accumulation task revealed that different regions displayed evidence tuning consistent with different candidate models. This work provides a mechanistic explanation for how graded information may be precisely accumulated within and transferred between neural populations, and suggests that different brain regions may accumulate evidence through different circuit mechanisms.

The accumulation of evidence for different alternatives is thought to be a fundamental cognitive operation. As such, neural recordings have been performed in many tasks designed to probe how evidence is accumulated to make a decision, including tactile[1,2], visual[3–11], olfactory[12], auditory[13,14], motor[15], and value discrimination tasks[16–20]. Experiments using these tasks have been conducted with the goal of understanding how the neural circuitry within and across different brain regions contributes to the accumulation of evidence, subsequent maintenance of this information in working memory, and ultimate commitment to a decision based on this evidence[21].

A major class of models proposed to describe this decision making process is the drift diffusion model[22–24], in which evidence is accumulated with a noisy drift process, and decisions are made based on when the resulting accumulation reaches some threshold. Many circuit-based models have been shown to be equivalent to the drift-diffusion model[25]. Such circuit-based models include mutual inhibition models[26], feedforward inhibition models[27], and pooled inhibition models[28]. These basic circuit models have been extended to explain a number of different decision-making experiments, with recordings in different regions, where neurons show persistent activity[29–34].

The mechanistic essence of the circuit-based models is the accumulation of evidence along a low-dimensional attractor representing the decision variable. Thus, these models predict that the activity of single neurons ramps up or down with evidence throughout a decision-making task[35,36]. Such ramping activity during decision-making tasks has been observed in a number of different brain

[1]Princeton Neuroscience Institute, Princeton University, Princeton, NJ, USA. [2]Howard Hughes Medical Institute, Chevy Chase, MD, USA. [3]Center for Neuroscience, Department of Neurobiology, Physiology and Behavior, University of California, Davis, Davis, CA, USA. [4]Department of Ophthalmology and Vision Science, University of California, Davis, Davis, CA, USA. [5]Present address: Department of Pharmacology, School of Medicine, University of Virginia, Charlottesville, VA, USA. [6]Present address: Department of Psychological and Brain Sciences, University of Delaware, Newark, DE, USA. ✉e-mail: lindseysbrown@princeton.edu; iwitten@princeton.edu; msgoldman@ucdavis.edu

regions in different paradigms, including in the lateral intraparietal area (LIP), medial temporal area (MT), and ventral intraparietal area (VIP) of monkeys in a random dot motion discrimination task[37–40], in the dorsal premotor cortex of monkeys during discrimination tasks[41,42], and in the posterior parietal cortex (PPC) and frontal orienting fields (FOF) of rats in an auditory evidence accumulation task[14,43]. These classes of neural circuit models that accumulate evidence through persistent, ramping activity successfully capture much observed experimental data.

However, a mounting body of evidence from large scale recordings during a range of decision-making tasks suggests that, in many decision-making contexts, neurons across cortical and subcortical regions are not persistently active. Instead, neurons fire transiently and sequentially in a choice-specific manner. This occurs in decision-making tasks both with navigation[44–49] and without navigation[50–58], including in evidence accumulation tasks[44,49]. These data challenge existing models of evidence accumulation by contradicting a core tenet of previous models: the presence of persistent, rather than sequential, neural activity.

For a model to accumulate evidence through sequential neural firing, it must perform two key computations. First, as in traditional models based on persistent activity, it must accumulate evidence received at each position. Second, unlike traditional models, it must transfer information between neurons at sequential positions.

Here, we develop two classes of neural circuit models that perform these computations. In each model class, the transfer of accumulated evidence between neurons encoding sequential positions is accomplished via a position-modulated gating signal. The key difference in the models is in their encoding of evidence, which is either in the relative amplitude of activity in two competing chains of neurons or in the network location of a stereotypically shaped, unimodal pattern (bump) of neuronal firing. We tested these differentiating evidence tuning predictions in a dataset of 14,247 imaged neurons, collected across 4 cortical and subcortical regions in mice performing an evidence accumulation task in virtual reality[59]. This analysis revealed that while seemingly similar choice-selective sequences are present in this task across these regions, neurons in different regions differ in their evidence tuning properties. In the neocortex, neuronal tuning curves mainly exhibit a monotonically increasing or decreasing encoding of evidence, characteristic of the competing chains model class, whereas in the hippocampus, neurons tend to have narrow, non-monotonic tuning curves, consistent with the bump model. Thus, our results reveal different encodings of accumulated evidence across brain regions and suggest that different regions could use distinct circuit mechanisms to form their evidence representations.

## Results

### Traditional integrator models fail to explain choice-selective sequences in the accumulating towers task

The models presented below are motivated by a large dataset ($n = 14,247$ neurons; $N = 26$ mice) of previously unpublished calcium imaging data (see "Methods") from anterior cingulate cortex (ACC) and dorsomedial striatum (DMS), as well as recently published imaging data from hippocampus (HPC)[49] and retrosplenial cortex (RSC)[44]. These data were all obtained during the same navigation-based, accumulation of evidence task in virtual reality (accumulating towers task, Fig. 1A). In this task[59] (see "Methods"), mice navigate a T-maze with Poisson counts of visual cues in the form of towers appearing to each side. After a delay region with no cues, the mouse reaches the arms of the maze and is rewarded for turning to the side with more cues (Fig. 1A). To make the correct decision, the mouse must integrate the number of cues and hold this information in working memory during the delay region. Mice are able to perform this task with good accuracy that increases with the magnitude of the difference between the number of cues to each side[59].

Across brain regions, neurons fired sequentially rather than persistently (Fig. 1B–E and Supplementary Fig. 1). Such sequences were present throughout the task, beginning in the cue region and continuing throughout the delay region. Moreover, many of the neurons in these sequences were choice-selective, such that neurons preferentially fired depending upon whether the animal will ultimately choose to turn left or right (Fig. 1B–E).

Previous models of evidence accumulation do not produce the observed choice-selective sequences. This includes traditional circuit models that neurally instantiate the drift diffusion model (Fig. 2A, B), as well as bump-based integrators (Fig. 2C, D) commonly used to accumulate head velocity signals into a representation of head direction. Further, canonical sequence models, such as synfire chains[60–62] and more recent models of replay sequences[63] do not accumulate evidence, while other sequence models accumulate external signals but are not choice-selective[64,65]. This motivates our construction of sequence-based evidence-accumulation models.

### Competing chains models

We first present a class of models consisting of two chains of neurons. In these models, the current position of the animal is represented by the location of elevated neural activity along the chains and the accumulated evidence is represented by the relative amplitude of firing of the two neurons corresponding to this location. Here, we consider a model with mutual inhibition between neurons at the same position in the chains (mutually inhibiting competing chains model, Fig. 3A), but other competing chains models are possible (see below, Supplementary Fig. 2 and Supplementary Text). The temporal dynamics of the firing rate $r_{i,L}$ of the $i$th neuron in the left ($L$) side chain are given by the equation

$$\frac{\mathrm{d}r_{i,L}}{\mathrm{d}t} = -ar_{i,L} + \left[ br_{i,L} + cr_{i-1,L} - er_{i,R} + f1_{\text{left}}(t) + P_i(t) - T \right]^+ \quad (1)$$

and an analogous equation describes the firing rates $r_{i,R}$ of neurons in the right ($R$) side chain. Here, $a$ is the exponential decay rate of activity in the absence of input, $b$ is the weight of self-excitation (Fig. 3A, green self-loops), $c$ is the strength of feedforward synaptic connections from the previous neuron (green feedforward connections), and $e$ is the strength of the inhibitory connection from the neuron at the same position in the opposite chain (brown connections). The external inputs to be integrated by the network consist of brief pulses of the form $1_{\text{left}}(t)$, which is defined to be 1 if there is a left cue within 0.5 cm of the current position and 0 otherwise, and project to the chain with synaptic strength $f$. $P_i(t)$ is the position-gating signal, and $T$ is the firing threshold. The position-gating signal $P_i(t)$ controls the progression of activity along the chain by raising the neuron's input above the firing threshold $T$, and is zero for all neurons except the pair of neurons (one from each chain) corresponding to the position of the animal in the environment. The firing rates of neurons not at the active position decay exponentially with time constant $1/a$. Firing rates and inputs during a portion of an example trial are shown in Fig. 3B. See "Methods" for complete details and parameterization of the model.

At a given position, this model has the same structure as the traditional mutual inhibition model[26] (Fig. 2A), and this motif is repeated across different positions to form two chains. This motif accumulates and maintains evidence through positive feedback within and between the two neural populations receiving the active position gating signal. Competition is mediated by mutually inhibitory positive feedback between the chains, such that increases of activity in one chain result in corresponding decreases of activity in the other chain. We note that the accumulation process saturates at an upper bound when neurons in the nondominant chain reach zero firing. This is because, when one chain is at zero firing rate, the disinhibitory component of positive feedback between the chains is lost, so that further increases in the

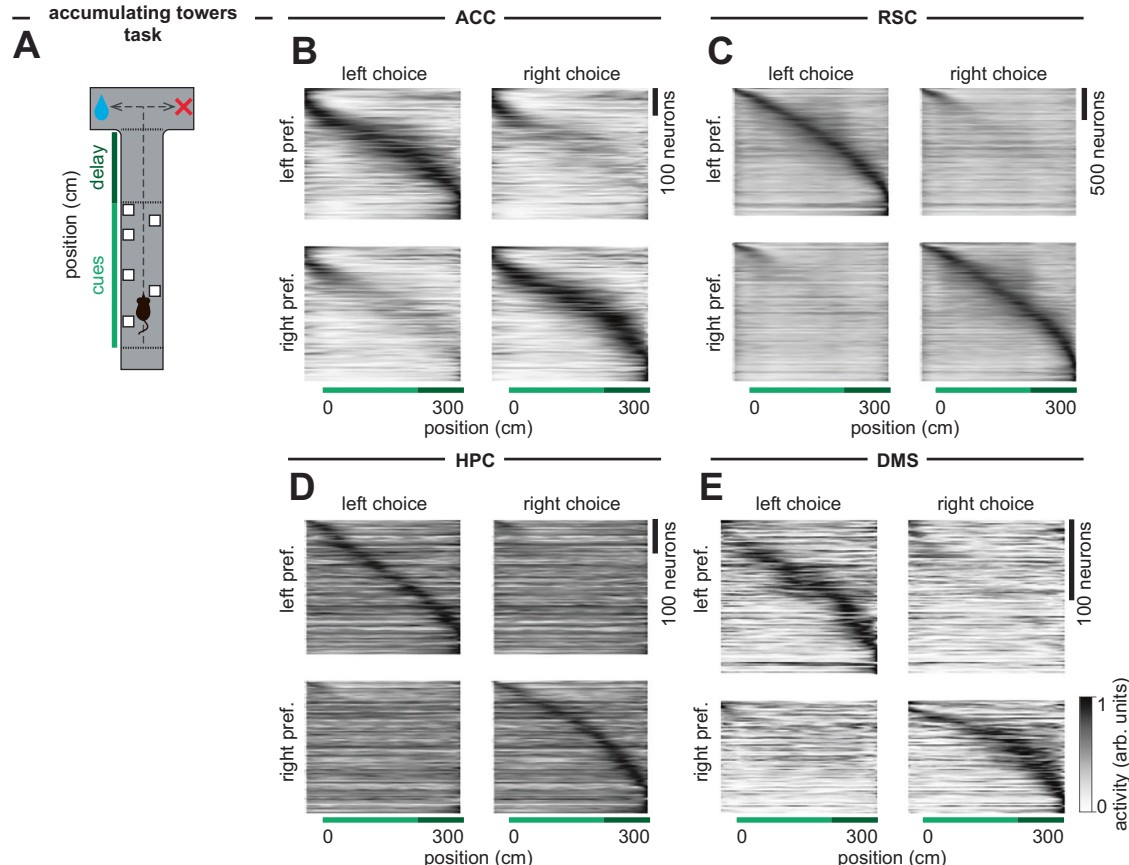

**Fig. 1 | Choice-selective sequences are observed across brain regions in a navigation-based, accumulation-of-evidence task. A** Schematic of a behavioral task in which mice navigate a T-maze in virtual reality. Visual cues (towers, white rectangles) are presented to either side of the maze during the cue period (light green, 200 cm long). At the end of the delay period (dark green, 100 cm long), mice are rewarded for turning to the side with more towers. **B** Each row shows the peak-normalized estimated firing rate at each position in the maze of a neuron with significant evidence tuning (see "Methods") recorded during this task from the anterior cingulate cortex (ACC, $n = 964$ neurons), averaged across trials when the animal turned left (left choice, left column) or right (right choice, right column). Neurons were assigned as left or right choice preferring based on their evidence-selectivity (see "Methods") and ordered based on the position of peak activity. **C** As in (**B**) but for retrosplenial cortex (RSC, $n = 4190$ neurons, data from Koay et al.[152]). **D** As in (**B**) but for hippocampus (HPC, $n = 791$ neurons, data from Nieh et al.[49]). **E** As in (**B**) but for dorsomedial striatum (DMS, $n = 322$ neurons). Sequences separated by even and odd trials are shown in Supplementary Fig. 1.

magnitude of evidence cannot be maintained. This saturation is not a core feature of the model, however, and the non-saturating range of evidence levels can be modified by the magnitude of external input to each chain or the synaptic strength of the visual input connections (Supplementary Fig. 2J and see "Methods"). As in the case of traditional, non-sequential accumulator models, perfect accumulation up to this upper bound requires fine-tuning of the parameters to prevent leak or uncontrollable growth in the integrator[35] (see Supplementary Text). Previous studies of such networks have also suggested that such accumulator networks may be made more robust to these fine-tuning conditions through incorporation of bistable components[66,67] or corrective feedback mechanisms[68,69].

Only the neurons receiving the position gating signal actively represent and accumulate evidence. When the animal moves to a new position, the position gating signal activates the next pair of neurons in the chain, and evidence is transferred to the newly activated pair of neurons by the feedforward synaptic connections[64] (see Supplementary Text for conditions on the feedforward connectivity for faithful transfer of evidence between positions). Repeating this process as the animal navigates down the track results in a sequence of neurons firing, with each successive pair of activated neurons representing the accumulated evidence in the difference of their firing rates.

We simulated the model on the accumulating towers task (see "Methods"), demonstrating that this model performs the task perfectly in the absence of noise for appropriately tuned parameters (Fig. 3C

and see Supplementary Text). Previous analysis of the accumulating towers task and other evidence accumulation tasks suggest that noise in an animal's performance is dominated by sensory noise, so that only a fraction of input cues are integrated[13,59]. When we correspondingly introduce input noise into our model (see "Methods"), the performance decreases in a manner similar to that observed in animals, with a more gradual increase in performance with increasing magnitude of the difference between cues (Fig. 3C). The neurons in the model exhibit choice-selective (or, more precisely, accumulated-evidence-selective) sequential activity, with neurons in each chain showing greater activity for the side of the evidence that it receives (Fig. 3D). Thus, accumulated evidence is encoded monotonically (up to saturation) in the relative firing rates of the two chains in this class of models (Fig. 3E).

Variant architectures of competing chains models are possible, including an uncoupled competing chains model where competition between opposing inputs occurs without inhibition between the chains (Supplementary Fig. 2A and see Supplementary Text). At a given position, this model has a push-pull feedforward arrangement of evidence inputs, with cues favoring one side exciting one chain and inhibiting the opposing chain, as in traditional decision-making models based on uncoupled competing accumulators[27]. This motif is then repeated to form two chains. As with the model based on mutually inhibiting chains of neurons, this uncoupled competing chains model is able to perform the task accurately (Supplementary Fig. 2B) and

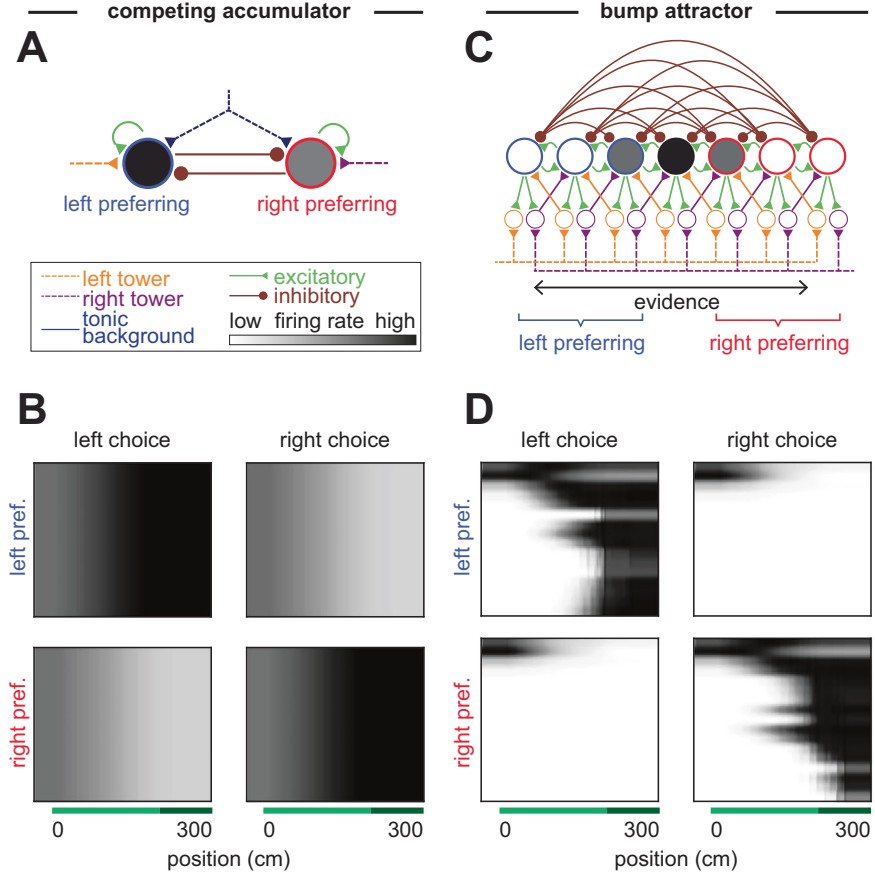

**Fig. 2 | Choice-selective sequences are not predicted by traditional integrator models. A** Schematic of a traditional mutually inhibitory integrator model for decision making. **B** Each row shows the average peak-normalized, simulated activity of a neuron in the model as a function of position in the maze, divided into the left and right population and sorted by the peak of mean firing activity. Each row is a neuron. **C**, **D** As in (**A**, **B**), but for a traditional bump attractor model. For model equations, see "Methods".

produce choice-selective sequences (Supplementary Fig. 2C) in which individual neurons have monotonic tuning curves to evidence (Supplementary Fig. 2D). As in the mutually inhibitory chains model, this model can either saturate or not depending on the strength of the synaptic input connections and assumptions about the dynamic range of the neurons (Supplementary Fig. 2K and see Supplementary Text).

Although these variant competing chains models cannot be reliably distinguished by their tuning since both models can be modified to have either linear or saturated tuning curves, the models instantiate different hypotheses that could be distinguished by functionally targeted optogenetic perturbations[70] (Fig. 3F and Supplementary Fig. 2E). We simulated the delivery of an excitatory input to a neuron (or similarly tuned neuronal population) while the animal is at the neuron's active position (Fig. 3F and Supplementary Fig. 2E) for a trial with no external cues so that the accumulated evidence prior to perturbation is zero. Due to the gating of neural activity by the forward-traveling position signal, both models predict that only neurons tuned to later positions will be affected by the optogenetic perturbation. For the mutually inhibiting chains model, subsequent neurons in the same chain will have increased activity due to this feedforward excitation, while subsequent neurons in the opposing chain will have decreased activity due to the mutual inhibition between the chains (Fig. 3G). By contrast, for the models based on uncoupled chains in which the anti-correlated activity of the two chains is solely due to opposing external cue inputs (Supplementary Fig. 2A), an excitatory perturbation in one chain increases firing only within the same chain but has no impact on the other chain (Supplementary Fig. 2F).

Another difference between the mutually inhibiting and uncoupled competing chains models is that the model based on mutual inhibition is more robust to the tuning of the position-gating. Specifically, the mutual inhibition model requires only that the position-gating signal $P_i$ be greater than or equal to the threshold $T$ at the active position (Supplementary Fig. 2G, H), whereas the uncoupled chains model requires that $P_i$ be exactly equal to $T$ to avoid continual growth of neural activity from temporally integrating the position-gating signal[71] (see Supplementary Text).

Taken together, these competing chains models represent a class of models in which position-gating gives rise to choice-selective sequences that encode graded evidence signals monotonically in the neuronal firing rates at each position.

**Position-gated bump attractors**

We next consider a class of models in which evidence is encoded in the location of a stereotypical, unimodal pattern (bump) of neural activity in the population. Such bump attractor models have been used to describe the neural circuitry that computes head direction in the rodent[72–77] and the fly[78–84], where the location of the bump corresponds to the animal's heading. Similarly, two-dimensional bump attractor models have been proposed to describe path integration in the hippocampal place cell system[85–87]. In these models, motion is temporally integrated to determine head direction or to perform path integration, and analogously, our models temporally integrate evidence from the visual cues. A traditional bump model that only accumulates evidence does not produce sequences like those observed in the neural data

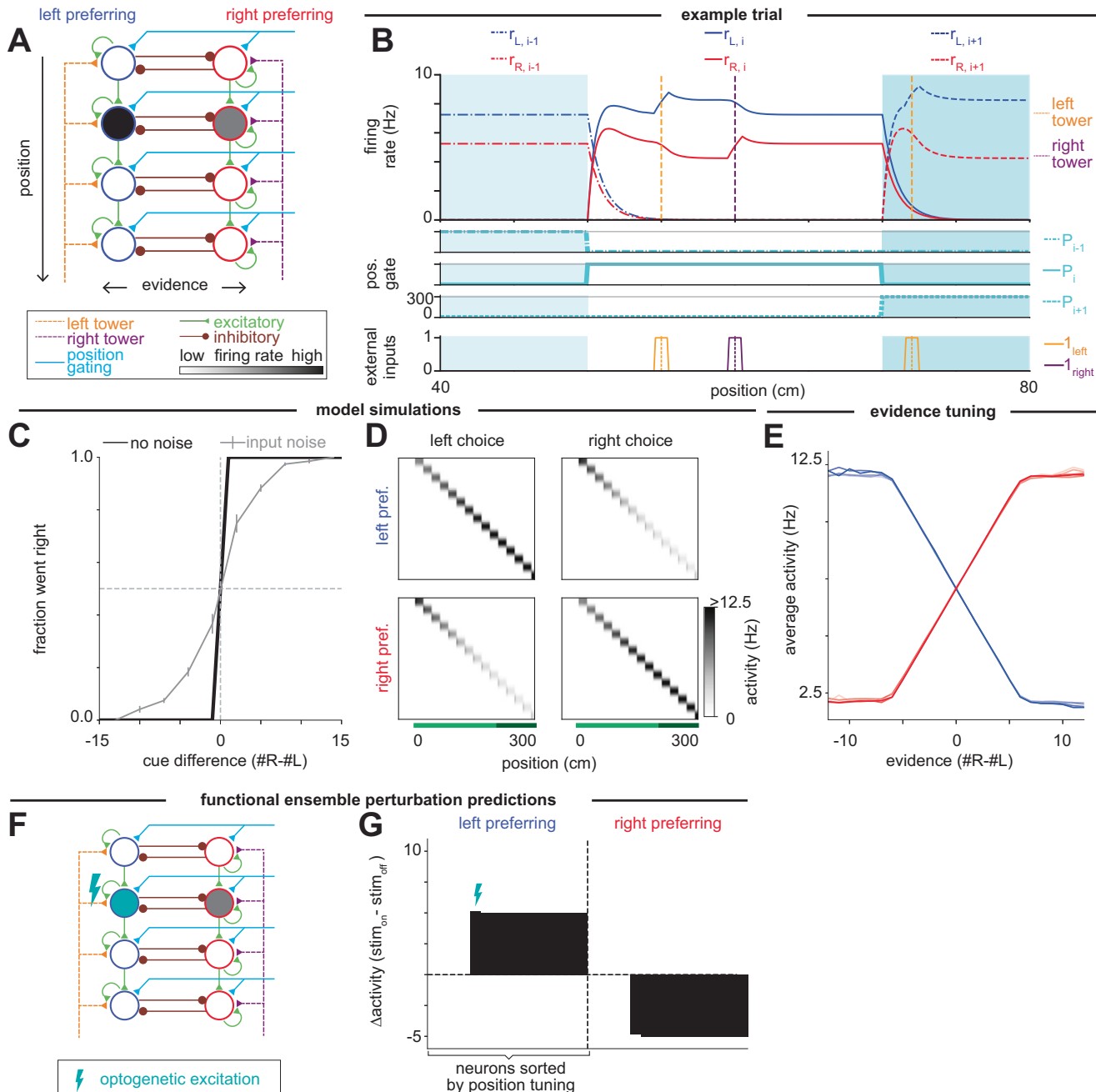

**Fig. 3 | Mutually inhibiting competing chains model of evidence accumulation through sequences. A** Schematic of neural circuit model for the mutually inhibiting competing chains model, showing excitatory (green) and inhibitory (brown) connections between neurons (circles) as well as external inputs to the circuit from the left (orange) and right (purple) towers and a position gating signal (cyan). **B** Top: Firing rates of neurons in the left (blue) and right (red) chains during an example trial, shown for positions 40–80 cm. Vertical dashed lines indicate locations at which left (orange) and right (purple) towers appeared. Background shading highlights regions where position gates $P_{i-1}$, $P_i$, and $P_{i+1}$ are on. Middle: Position gating signal at each position. Bottom: Input current to the left (orange) and right (purple) chains resulting from the external inputs. **C** Psychometric curve for the model, describing how often the amplitude of the final neuron in the right chain exceeded that of the final neuron in the left chain for model simulations with (gray) and without (black) noise in the input. Error bars indicate s.e.m. **D** Each row shows the non-normalized amplitude of a model neuron at each position in the maze, averaged across simulated trials without noise, when the greater final amplitude was in the left (left choice, left column) or right (right choice, right column) chain. Neurons were assigned as left- or right-preferring based on their choice-selectivity (i.e., which chain they belong to) and ordered based on the position of peak activity (i.e., their position in the chain). **E** Average firing of individual left-preferring (blues) and right-preferring (reds) neurons as a function of evidence when the neurons are activated by the position-gating signal. **F** Schematic of a single-neuron perturbation experiment where optogenetic excitation is applied to a single neuron in the left chain. **G** Simulated changes of the firing rates of all neurons in the absence of cues when a single neuron (denoted with the lightning bolt) is optogenetically stimulated.

(Fig. 2D) because the bump would recur at the same location whenever the accumulated evidence is at the same level and would remain at a fixed location throughout the delay period when the evidence level does not change. Thus, we modify these traditional bump attractor models to have separate dimensions for encoding position and evidence (Fig. 4 and Supplementary Fig. 3).

The model consists of layers of recurrently connected neurons, with each layer corresponding to a given position and neurons within

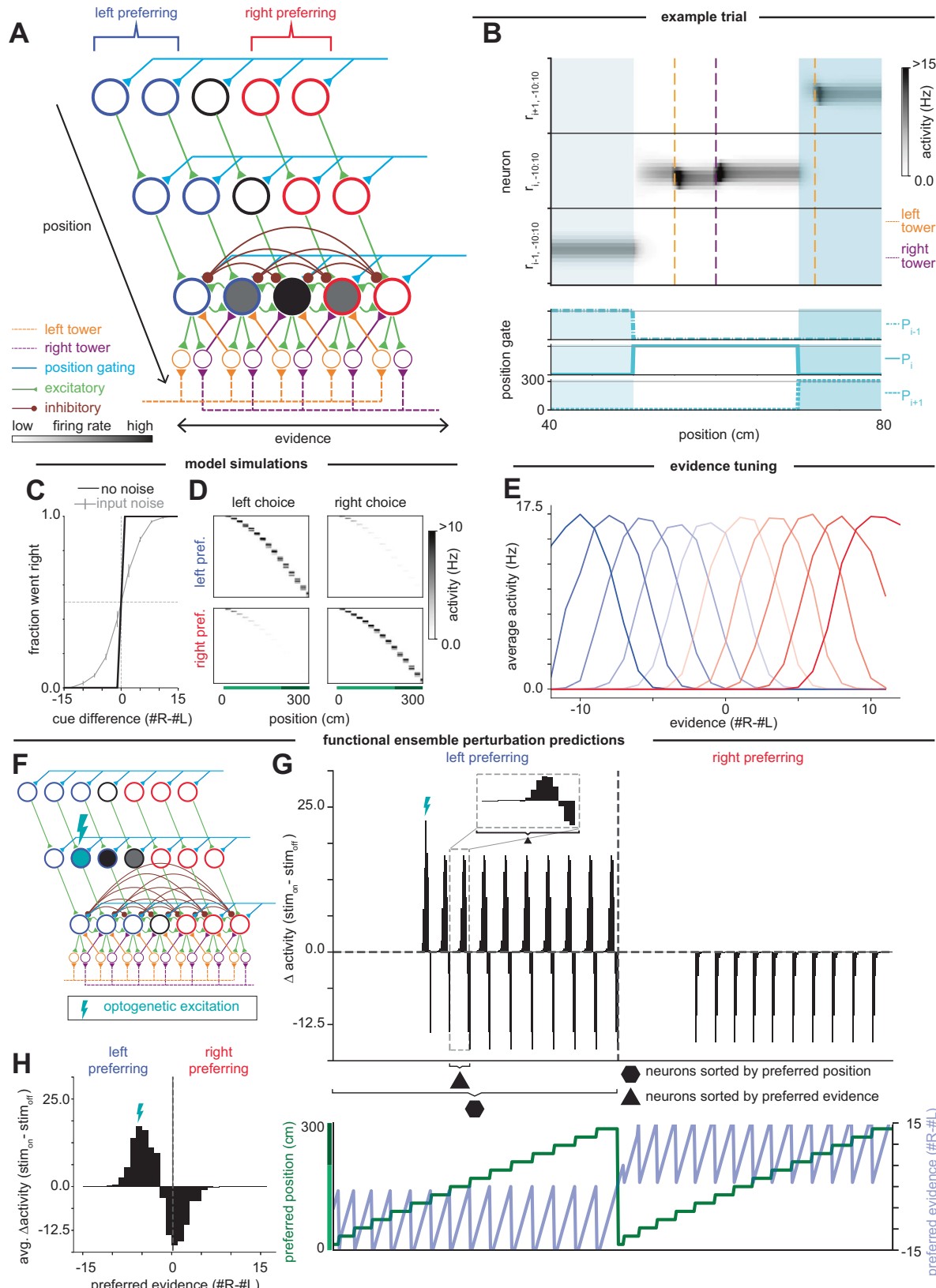

the layers representing the accumulated evidence at this position. The activity $r_{i,j}$ of a neuron at position $i$ and evidence level $j$ evolves according to

$$\frac{dr_{i,j}}{dt} = -ar_{i,j} + F\left(\sum_k W_{j,k} r_{i,k} + br_{i-1,j} + c\left(I_{i,j+1,L}(t) + I_{i,j-1,R}(t)\right) + P_i(t) - T\right)$$

(2)

where $a$ is the exponential decay rate of activity in the absence of input, $P_i(t)$ is the position-gating signal that selects the active layer of neurons, and $T$ sets the soft threshold of the $(1 + \tanh(\cdot))$ neuronal output nonlinearity $F$. Within a layer, $W_{j,k}$ is the strength of the connection from the neuron representing evidence level $k$ onto the neuron encoding evidence level $j$. $W_{j,k}$ is excitatory for connections of a neuron onto itself and its immediate neighbors (Fig. 4A, green

**Fig. 4 | Position-gated bump attractor model of evidence accumulation through sequences. A** Schematic of neural circuit model for the position-gated bump attractor model. Each row represents neurons (circles) that respond at a given position. Within a row, each neuron represents a different evidence level, ranging from left-most to right-most. Blue: left-preferring neurons; red: right-preferring neurons. Activation of neurons encoding a given position is controlled by a position gating signal (cyan). Transfer of information between positions is controlled by excitatory feedforward connections between positions (green). Bottom row of neurons illustrates the recurrent (green: excitation; brown: inhibition) and cue-related (orange: deriving from left towers, purple: deriving from right towers) inputs received by the neurons encoding a given position. Left and right tower cues enter the network via corresponding shifter neurons (orange and purple circles). **B** Top: Firing rates of a subset of neurons (rows, where $r_{i,j}$ is the firing rate of the neuron at position $i$ and evidence level $j$) during an example trial, shown for positions 40–80 cm. Neurons are first sorted by position tuning (black horizontal lines divide positions), then by evidence tuning within a position. Dashed vertical lines indicate positions at which left (orange) and right (purple) cues appeared. Shading highlights regions where position gates $P_{i-1}$, $P_i$, and $P_{i+1}$ are on. Bottom: Position gating signal at each position. **C, D** Psychometric curves and trial-averaged firing rates of neurons at each position in the maze, as in Fig. 3C, D, but for the position-gated bump attractor model. **E** Tuning curves of a subset of left-preferring (blues) and right-preferring (reds) neurons to evidence, calculated when the neurons are activated by the position-gating signal. **F** Schematic of single-neuron perturbation experiment in which optogenetic excitation is applied to a single left-preferring neuron. **G** Top: Simulated changes of firing rates of all neurons in the absence of cues when a single neuron (denoted with lightning bolt) is stimulated. Neurons are first sorted by left- or right-preferring, then by position tuning, and then by evidence tuning within a position. Bottom: Preferred position (green) and evidence (light blue) tuning of each neuron in the top panel. **H** Average change in firing rate for neurons tuned to different evidence levels.

recurrent connections) and inhibitory onto all other neurons in that layer (Fig. 4A, brown recurrent connections). The coefficient $b$ is the synaptic strength of feedforward excitatory connections between layers of neurons encoding neighboring positions, and we assume for simplicity that these connections are only between neurons encoding the same evidence level (Fig. 4A, green feedforward connections). The terms $I_{i,j+1,L}$ and $I_{i,j-1,R}$ are the firing rates of the shifter neurons (analogous to the rotation cells in Skaggs et al.[72]) (Fig. 4A, orange and purple circles) that shift the bump left or right in response to external cue inputs (Fig. 4A, left and right tower inputs; Fig. 4B, shift in location of the bump following an input). The left shifter (orange) neuron is activated when jointly receiving input from evidence accumulating neuron $j+1$ and a left-side external cue. In this manner, if the peak of the bump of neuronal activity was previously at $j+1$, this peak activity then (through a synaptic connection of strength $c$ from the shifter neuron) drives the neuron at evidence level $j$. Mathematically, the firing rates of the left, and corresponding right, shifter neurons are given by

$$I_{i,j,L} = r_{i,j}1_{\text{left}}(t) \tag{3}$$

$$I_{i,j,R} = r_{i,j}1_{\text{right}}(t) \tag{4}$$

where $1_{\text{left/right}}(t)$ equals 1 if there is a left or right cue, respectively, within 0.5 cm of the current position. For the complete model specification, see "Methods".

Within the active position, accumulated evidence is represented and maintained in the location of a bump of activity along the evidence axis. As in traditional bump attractor models[88–90] (Fig. 2C), the shape of the bump is maintained in the absence of input by local symmetric excitation and long range inhibition. Asymmetric input connections cause the bump to shift left or right in response to a left or right cue, respectively[73,77], as described above (see Supplementary Text). This shift of the localized bump of activity occurs despite the external cue signal being global (Fig. 4A, tower inputs). Mechanistically, the ability to shift a localized bump of activity with a global external signal is achieved via the multiplicative gating of the external input by the activity of the evidence-encoding neurons (Eqs. 3 and 4) so that the global external input signal only activates the shifter neurons close to the location of the current bump. Alternatively, this gating could be accomplished with additive inputs plus thresholds[84,88].

As the animal progresses forward in the maze, the active position changes so that information must be transferred to the next layer of neurons, requiring both a change in the position gate and feedforward connections between layers (Fig. 4B). Here, position-gating is controlled by the position signal $P_i(t)$ and the soft threshold $T$, so that the output of $F$ is only significantly greater than zero when the position signal is active (i.e., $P_i(t) \geq T$). Away from the active position (i.e., where

$P_i(t) = 0$), activity decays exponentially with time constant $1/a$ (Fig. 4B). Feedforward excitation between the layers then transfers information to neurons at the next position. We have modeled these feedforward connections as only projecting between neurons representing the same evidence level, but this projection could be broader as long as the feedforward input is symmetric and centered around the current evidence level (see Supplementary Text). As in the chains-based models, we have presented the position signal as an external input. However, position could simultaneously be generated within the network through an integration of velocity inputs, by including a similar arrangement of excitatory and inhibitory connectivity along the position axis. In this case, the network forms a planar bump attractor that jointly encodes both evidence and position (Supplementary Fig. 3 and see Supplementary Text).

We simulated the position-gated bump attractor model for the same set of trials as the competing chains models. As in the competing chains models, for appropriately tuned parameters, the position-gated bump attractor also performs the task perfectly in the absence of noise, with performance worsening when input noise is introduced (Fig. 4C). The simulated neural activity also produces choice-selective sequences (Fig. 4D). As seen by the downward curvature of the sequences in Fig. 4D, the sequences are marked by the feature that more neurons respond at late positions than respond at early positions in the sequence. This is because the random walk of the bump location as evidence is accumulated leads to a greater range of evidence levels being observed at later positions in the trial; thus, on average across trials, there are more different neurons active at later positions, with each neuron corresponding to a different possible evidence level. However, we note that this is not a fundamental prediction of the model because, experimentally, the exact shape of the choice-selective sequence depends on the number of neurons residing within each uniformly spaced evidence-position location in the model (Fig. 4A). We also note that our assumption of perfect symmetry simplifies the analysis, but is not required to produce bump attractor models[91,92] and that the robustness of bump attractor models to diffusion and drift of the bump in the presence of noise has been extensively studied[91,93–95] and may be reduced through error-correcting codes[96], extending representations to multiple bumps[97], incorporation of bistable components[93,98], or relaxing the continuous attractor assumption to have only a discrete set of memory states[99].

The most straightforward difference between the position-gated bump attractor and the competing chains models is in the shape of their evidence tuning curves. Due to the structure of the network connectivity, the neurons in the position-gated bump attractor are non-monotonically tuned to specific evidence levels at their active position (Fig. 4E), in contrast to the monotonic tuning curves of the competing chains models (Fig. 3E).

To further differentiate the position-gated bump attractor model from the competing chains models, we consider predictions for the

effects of optogenetic excitation of a single neuron during a trial with no cues presented (Fig. 4F). Here, we consider the regime where the stimulation is sufficiently large to cause the bump to move to the stimulated location. Due to the similar feedforward connectivity patterns between positions in both model classes, stimulation of a neuron active at one position only affects the activity of neurons tuned to later positions, as in the competing chains models. However, unlike the competing chains models, we see that rather than impacting every neuron at a later position equally, the effects are sparse and differ in sign and magnitude across neurons (Fig. 4G). These differences are because the stimulation has caused the bump to move to a new evidence level, which is then sustained at subsequent positions. Thus, neurons with a similar evidence tuning to the neuron that was stimulated increase in activity, while those tuned to where the bump of activity would be located in the absence of stimulation (at 0 evidence) decrease their activity since the bump has moved to the new location (Fig. 4H). We note that for smaller excitatory stimuli that do not shift the bump center to the site of stimulation, and dependent upon the exact details of the excitatory and inhibitory connections, smaller shifts in the bump either toward or away from the stimulated location are possible[95,100–102], but the perturbations still maintain the properties that changes are sparse and that changes within the same choice-preferring population as the stimulated neuron may be bidirectional.

## Neural activity in different brain regions correlates with different classes of models

Above, we presented two circuit models for accumulating evidence through sequences (competing chains models: Fig. 3 and Supplementary Fig. 2; position-gated bump attractor model: Fig. 4 and Supplementary Fig. 3). While both models produce neurons that are sequentially tuned to position, the models predict different evidence coding schemes. The competing chains models make two core predictions: first, evidence should be monotonically encoded in neuronal activity and, second, the monotonic encoding within the population should be graded across a range of evidence levels. The bump attractor model likewise makes two core predictions: first, evidence should be encoded with relatively narrow, unimodal tuning and, second, the peaks of these tuning curves should be broadly distributed across the entire evidence range. We looked for signatures of these different evidence encoding schemes in recordings from ACC, RSC, DMS, and HPC during the accumulating towers task (Fig. 1A).

Towards this end, we first identified cells that were evidence-tuned, showing any consistent response pattern at different accumulated evidence values. For these evidence-tuned cells, we then identified whether neurons were monotonic-like or unimodal-like in their evidence tuning and also, for unimodal tuning, we characterized whether the peaks of the tuning curves were broadly distributed across the range of evidence levels (Supplementary Fig. 4). Within the class of monotonically evidence-tuned neurons, we next assessed whether the evidence tuning curves had broadly graded tuning to evidence, versus more binary tuning that would be consistent with a choice-like representation, and also how this distribution varied with position in the maze. We further examined whether graded evidence could be decoded from the population as a whole and whether the encoding of evidence across the population was consistent with the prolonged, temporal response kernels required for graded accumulation of evidence.

## ACC and RSC neurons primarily have broad monotonic evidence tuning, while HPC neurons have narrow unimodal evidence tuning

Identifying neurons that were tuned to evidence requires jointly, rather than independently, fitting position and evidence to account for the correlated, nonuniform sampling of these two variables (Supplementary Fig. 4A, B). For example, neurons tuned to early positions in

the maze are only active when the accumulated evidence is small, given that little evidence has been presented at early positions. Thus, if such neurons were directly fit to evidence tuning alone, rather than jointly fit to evidence and position, they would appear to be tuned to small absolute evidence levels even if they had no tuning to evidence at all. Likewise, because the evidence levels sampled at later parts of the maze were most concentrated around moderate-to-large absolute values (Supplementary Fig. 4A), neurons untuned to evidence but tuned to later positions in the maze would artifactually appear to have a bimodal tuning to evidence.

To capture many different tuning curve shapes across the population, we first modeled the joint tuning for position and evidence as a product of a Gaussian function of position and a Gaussian function of evidence (Supplementary Figs. 4B, 5C, D and Fig. 5C, D, G, H, K, L; see "Methods"). The Gaussian in position intuitively captures the non-monotonic, sequential position tuning. The Gaussian in evidence, although non-monotonic in its mathematical form, allowed us to identify not only non-monotonic but also monotonic evidence tuning, because in the latter case the best fit Gaussian has a peak near the extreme of the evidence range (i.e., normalized peaks near $\pm 1$; Supplementary Fig. 4D). Broad monotonic ramps like those of the competing chains models also have a relatively large standard deviation relative to the observed evidence range (Supplementary Fig. 4E, F).

Jointly fitting position and evidence tuning on correct trials across the 14,247 neurons revealed that 56% of all neurons in ACC, 48% in RSC, 25% in HPC, and 41% in DMS had significant evidence tuning (showing any pattern of response to evidence with statistical comparison to pseudosessions; see "Methods"). In analyzing neurons with significant evidence tuning in each region, we saw distinguishing patterns emerge in the nature of the evidence tuning (Fig. 5 and Supplementary Fig. 5).

In ACC, many individual neurons exhibited primarily monotonic and relatively broad evidence tuning (Fig. 5A, B and Supplementary Figs. 6A, 7). To accurately visualize evidence tuning, in addition to plotting raw position and evidence fields (Fig. 5A), we also fit both a Gaussian and a logistic function to the peak position tuning (Supplementary Fig. 4C) for a number of example neurons, and plotted the best fit function (Fig. 5B, F, J). Similar to these example neurons, among the population of significantly evidence-tuned neurons, peak evidence tuning was towards the extreme of the observed evidence range, consistent with monotonic tuning (Fig. 5C, D). The presence of monotonic tuning is most consistent with predictions of the competing chains models (Fig. 3E and Supplementary Figs. 2D, J, K, 4E, F), although the logistic fits revealed many neurons had steeper sigmoidal tuning than produced by the competing chains models (Fig. 3E and Supplementary Fig. 2D, J, K), especially at later positions in the maze (Fig. 5B and Supplementary Fig. 6A). The steepness of such sigmoidal tuning, and its variation across different positions of the maze, is treated further in the next section.

In contrast, HPC neurons showed primarily non-monotonic and narrow evidence tuning (Fig. 5E, F and Supplementary Figs. 6B, 8), which tiled the range of observed evidence (Fig. 5F–H). These narrow tuning curves occurred throughout the cue and delay region (dark blue neurons in Fig. 5G). The presence of non-monotonic, narrow tuning curves that tile evidence space is consistent with predictions of the bump model (Fig. 4E and Supplementary Fig. 4G, H). In addition to narrow tuning curves, the bump model also predicts that the population should track the accumulated evidence on individual trials. Such narrow tuning curves make decoding challenging because, to perfectly decode evidence, recordings would need to include cells tuned to each combination of position, evidence, and other variables previously identified in this task[49], and the HPC recordings are sparse. However, previous analysis of these data suggests that evidence is represented in the population on a nonlinear neural manifold in HPC[49], and we similarly find that evidence can be decoded above chance using another nonlinear embedding technique[103] (Supplementary Fig. 11).

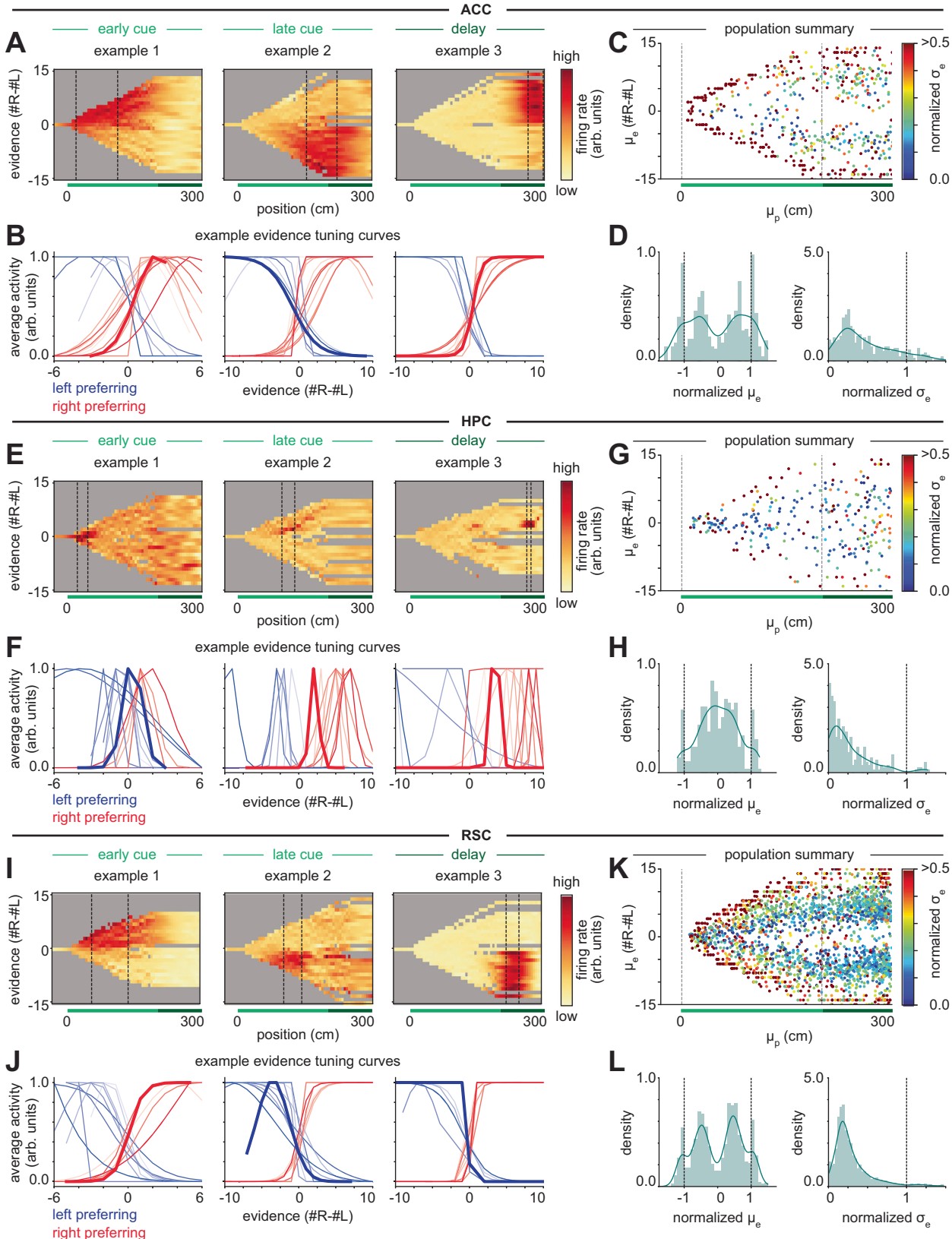

In RSC, the majority of neurons showed monotonic patterns (Fig. 5I, J and Supplementary Figs. 6C, 9) like those observed in ACC, while a smaller population showed more narrow tuning curves like, but typically somewhat wider than, those observed in HPC (example 2 in Fig. 5I and examples 3, 9, 16, 22, 28, and 34 in Supplementary Fig. 9). These observations are seen in the population summary of evidence and

position tuning (Fig. 5K, L), which shows that many neurons have wide evidence tuning with peaks at the extreme (deep red) while many others have narrow intermediate evidence tuning (dark blue). RSC also has a particularly large population of neurons that, when fit to a Gaussian, have intermediate widths and evidence peaks not fully at the

**Fig. 5 | ACC and RSC neurons primarily have broad monotonic evidence tuning, while HPC neurons have narrow unimodal evidence tuning. A** Heatmaps showing average firing in position-by-evidence bins for example ACC neurons with peak position tuning in the early cue (left), late cue (middle), or delay (right) region of the maze. Gray denotes bins with fewer than 2 samples during the session. **B** Example ACC neuron evidence tuning curves fit to the position of the neuron's peak activity (observations within 0.5 position standard deviations of the neuron's fit peak position tuning) for a collection of neurons with mean position tuning in the early cue (left), late cue (middle), or delay (right) region of the maze. Red: right-preferring neurons. Blue: left-preferring neurons. Bold lines correspond to the examples in (**A**), for which the neuron's region of peak activity is the region between the dashed vertical lines. For comparison, raw evidence tuning curves are provided

in Supplementary Fig. 6. **C** Scatter plot showing the location of the fit mean position ($\mu_p$) and mean evidence ($\mu_e$), with color indicating the normalized fit evidence standard deviation ($\sigma_e$) (see "Methods") of the 80% of neurons recorded in ACC with the best fit between the neural data and model predictions. Narrowly tuned neurons are only plotted if they pass non-outlier criteria (see "Methods"). Dashed lines indicate boundaries of the cue period. **D** Density plots of normalized fit mean evidence (left) and normalized fit evidence standard deviation (right) for the neurons shown in (**C**). Solid blue lines show the kernel density estimate. **E**–**H** Same as (**A**–**D**) but for neurons recorded in HPC. **I**–**L** Same as (**A**–**D**) but for neurons recorded in RSC. Distributions for all evidence-modulated neurons (regardless of goodness-of-fit and outlier tests) are shown in Supplementary Fig. 12.

extremes (Fig. 5K, green); such neurons typically exhibit a steep sigmoidal tuning curve, such as that shown in example 3.

DMS was dominated by monotonic evidence tuning, but with some narrowly tuned neurons (Supplementary Figs. 5, 6D, and 10). However, DMS contained relatively few evidence-modulated neurons with peak position tuning in the cue region. Instead, most evidence-modulated neurons had their peak position tuning in the delay region and had choice-like steep sigmoidal evidence tuning (Supplementary Fig. 5C, E).

### Shift from evidence tuning in the cue period to choice tuning in the delay period in ACC and RSC

While the monotonicity of the evidence tuning of individual cells observed in ACC and RSC is consistent with the competing chains models, a candidate integrator must represent graded evidence beyond monotonic, step-like evidence tuning driven by choice. Since the units of our models should not be thought of as individual cells, but rather populations of neurons with similar position tuning, the graded representation predicted within our model units could emerge from either individual cells having graded representations or individual cells stepping on at different evidence thresholds such that population averages are graded[104]. Therefore, we examined whether the regions dominated by monotonic tuning for evidence contained representations of evidence in the population beyond binary choice tuning, using both single-cell and population-level analyses.

We first asked whether there were signatures of evidence accumulation within the population by deriving cue response kernels from a linear cue encoding model fit to the difference in right- and left-preferring population activity (Fig. 6A–C and see "Methods"). As required for integration, we found cue-locked, sustained, and relatively constant steplike responses to the cues, with left and right cues having a similar pattern but with opposite signs (Fig. 6D, E). When accounting for the amount of evidence accumulated so far, the magnitude of the cue responses decreased, consistent with a saturation at more extreme evidence levels and a more choice-like representation as the maze progresses (Supplementary Fig. 13A, B). Interestingly, the cue response kernels decreased less as a function of position in the maze than as a function of accumulated evidence (Supplementary Fig. 13C, D; note nonzero kernels even for later positions), suggesting that the transition observed as a function of previously accumulated evidence may be due to animals committing to a decision with larger levels of evidence[105,106], rather than solely being due to being at a later position in the maze. Most importantly, this analysis suggests that accumulated evidence is represented beyond choice at the population level (at least for intermediate values of evidence).

We next sought to quantify the mixture of graded evidence and binary choice activity in individual cells. To do so, we fit single-neuron evidence vs. choice encoding models at each position bin, using evidence and behavioral choice as regressors for both correct and incorrect trials (Fig. 6F and see "Methods"). At each position, we calculated the fraction of active neurons (see "Methods") whose firing rate was significantly modulated by the linear evidence or the binary

choice variable, while accounting for the contribution to neural activity of the other variable. In both ACC and RSC, we observed neurons with significant encoding of evidence beyond that explained by choice during the cue period, with decreasing proportions of cells encoding evidence during the delay period (crimson traces, Fig. 6G, H). In contrast, the fraction of significant choice encoding neurons (beyond that explained by the evidence) increases throughout the maze, with large fractions of active neurons encoding choice in the delay period (lime traces, Fig. 6G, H). The above analysis primarily differentiates behavioral choice-tuned neurons from evidence-tuned neurons by their change in tuning to evidence between correct and incorrect trials. However, a subtlety is that a neuron that has binary tuning to the current best choice (i.e., the current sign of the evidence), rather than graded tuning to evidence, may be identified as evidence tuned despite not being graded in its response. Therefore, to test whether there was graded response in the population of cells identified as evidence-tuned, we plotted the trial-averaged firing of the active cells in this subpopulation and confirmed that this subpopulation exhibited graded patterns of activity with evidence during the cue period, consistent with our model predictions for accumulation (Supplementary Fig. 13E, F).

To complement this single-neuron encoding analysis, we also asked whether evidence could be linearly decoded from the population (Fig. 6I). Similar to the findings from the single-neuron analysis, evidence can be linearly decoded from the population of active cells above chance early in the maze, falling to chance levels in the delay region (Fig. 6J, K). This is true even when controlling for the sign-of-evidence and considering only nonzero evidence levels to eliminate the possibility of a binary encoding of just the current sign-of-evidence (Supplementary Fig. 13G, H). Together, these results imply that graded evidence representations in the cue region transition to a more binary choice representation in the delay period.

We sought to summarize these results with a population level visualization. The population-averaged position-evidence responses in both ACC and RSC show graded increases in activity with preferred-evidence (i.e., evidence calculated in reference to each neuron's preferred direction of evidence) during the early portion of the cue period (Fig. 6L, M, dark blue traces; population responses based on averaging all evidence-modulated neurons, see "Methods"). Consistent with the other analyses (Fig. 6G, H, J, K), with progression along the maze, this evidence tuning sharpens, transitioning to a step function by the start of the delay period (magenta and purple traces) that resembles tuning to choice rather than graded evidence tuning. In ACC (and more subtly in RSC), population responses also increase later in the maze, due to the fact that there are more evidence-tuned neurons in the delay region that contribute to these population averages (Supplementary Fig. 12A, B, left panels).

### Augmenting the circuit models to recapitulate the transition from graded evidence to binary choice tuning

Neither of the competing chains models produce this transition from relatively more evidence-encoding to relatively more choice-encoding,

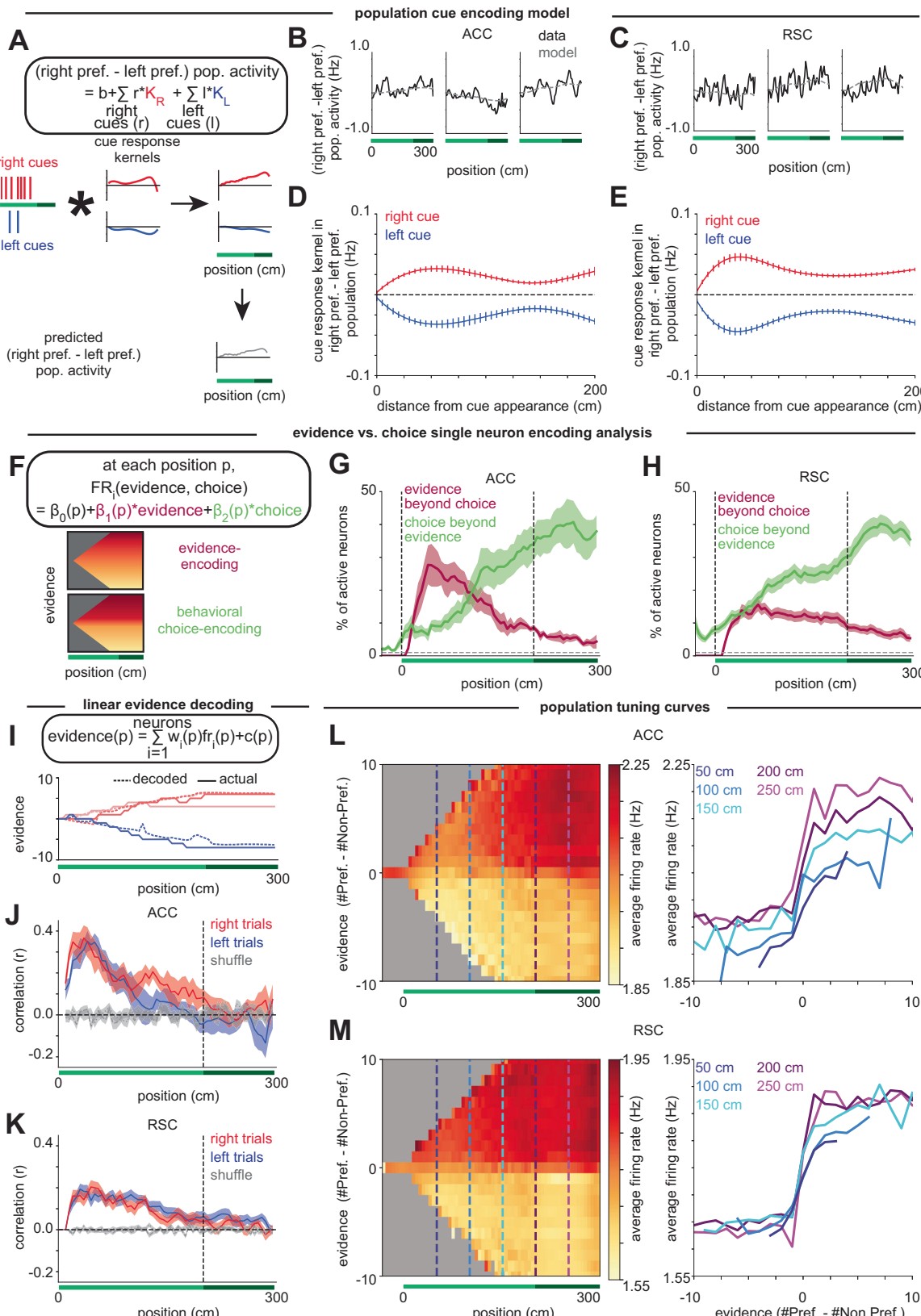

since the models were designed to perfectly accumulate evidence. This transition could be realized in the models in multiple manners. First, a population of choice readout cells at each position could compare the accumulation of the left and right chains at that position and respond in a binary fashion when either the left or right chain has greater activity, where the number of readout cells at each position increases

over the course of the maze (Fig. 7A–C). Second, this pattern could be realized if the parameters of the mutually inhibiting chains model are slightly mistuned such that the integrator becomes increasingly unstable later in the maze (Fig. 7D–F). Such slight instability has been suggested in previous integrator models, where the instability has been functionally interpreted to provide an urgency signal that

**Fig. 6 | In ACC and RSC, evidence tuning in the cue period shifts to choice tuning in the delay period. A** Schematic of the population cue encoding model, where each left and right cue is convolved with a left or right cue response kernel, $K_L$ or $K_R$, respectively, to predict population activity. **B** Example single-trial population activity (black) compared to model predicted activity (dashed gray) in ACC. (Average $r^2 = 0.26$ across 7 sessions) **C** Same as (**B**) but for RSC. (Average $r^2 = 0.20$ across 41 sessions). **D** Cue kernels to right (red) and left (blue) cues in ACC. Error bars indicate s.e.m. across sessions. **E** Same as (**D**) but for RSC. **F** Evidence vs. choice single-neuron encoding analysis. Top: Regression equation. Bottom: Schematics of firing for a purely evidence encoding cell and a purely choice encoding cell on correct trials. **G** For ACC, fraction of the active neurons at each position (see "Methods") with significant coefficients for evidence (crimson) or choice (lime) in the choice vs. evidence encoding model, based on the F-statistic. Lines and shading

indicate mean ± s.e.m. across sessions. The horizontal dashed line indicates the chance level (1%). **H** Same as (**G**) but for RSC. **I** Top: Linear evidence decoding model at each position. Bottom: Example trials with actual (solid) and decoded (dashed) evidence in ACC. **J** At each position, the cross-validated correlation between actual evidence and decoder-predicted evidence in ACC on correct right evidence ($e \geq 0$; red) and left evidence ($e \leq 0$; blue) trials, compared to shuffle (gray). Shading indicates s.e.m. across sessions. **K** Same as (**J**) but for RSC. **L** Left: Heatmap showing average firing rates across evidence-tuned neurons in ACC in bins of position by preferred-evidence. Gray indicates bins that were not sampled more than twice on at least 10% of the sessions. Right: Cross-sections of heatmap at points in the cue period (50 cm, dark blue; 100 cm, medium blue; 150 cm, cyan) and delay period (200 cm, purple; 250 cm, magenta), showing average firing rate across neurons as a function of preferred evidence. **M** Same as (**L**) but for RSC.

accelerates decision commitment[107,108]. These mechanisms are not mutually exclusive. Thus, the prevalence of choice cells in the data complements the integration processes focused upon in our models by suggesting that there are additional processes at play driving the transition from evidence accumulation to choice representation.

## Discussion

This work proposes mechanistic models for the widespread observation of choice-selective sequences during evidence accumulation (Fig. 1). Since traditional models of evidence accumulation based on persistent activity fail to explain choice-selective sequences (Fig. 2), we present two classes of circuit models: competing chains models (Fig. 3) and position-gated bump attractor models (Fig. 4). These two model classes represent two different evidence accumulation schemes, in which evidence is encoded either in the amplitude of the neuronal firing rates or in the set of neurons that fire with stereotyped amplitude, and predict different effects of targeted perturbations. We tested the evidence tuning predictions in imaging data across multiple cortical and subcortical brain regions, demonstrating that although different regions show seemingly similar choice-selective sequences, their underlying evidence representations are distinct (Figs. 5 and 6). Specifically, neocortical regions (ACC and RSC) had primarily monotonic evidence tuning curves in the early cue region, characteristic of the competing chains models, while the HPC showed non-monotonic evidence tuning with broadly distributed tuning curve peaks throughout the maze, consistent with the position-gated bump attractor.

### Transfer of accumulated evidence across positions is a fundamental cognitive operation

Although our models were tested on data from a specific task, at the essence of accumulating evidence through sequences is the ability to both integrate evidence and transfer information between different populations of neurons. These two operations apply quite generally to many decision-making contexts. For example, position in our models should be interpreted broadly as a sequentially progressing state of the animal (e.g., it could represent time), so these models apply to navigation or non-navigation tasks. Moreover, perceptual decision making tasks may require transferring accumulated evidence across neuronal populations when the population that needs this information to carry out an action changes (e.g., changing the visual gaze position while maintaining evidence for two saccadic choice targets[109]). While models for evidence integration have been explored extensively and form the core of our models' computations within a position, traditional models do not transfer graded information between positions, a core feature of our models.

To achieve the transfer of information between positions, our models use a network architecture in which neurons project synaptically to neurons encoding the subsequent position, but these connections are only effective in transferring information when a position-

gating signal pushes the receiving neurons above their firing threshold. While not explicitly modeled here, such spatio-temporal gating could result from inputs from the basal ganglia[110–113], place cells in the hippocampus, or cells within each region that show position but not evidence tuning[44,49] (Supplementary Fig. 1). The position gating signal in our model may itself result from an integration process, as in previous models of hippocampal path integration[85–87], with the position integration performed in a separate population (Supplementary Fig. 2I) or within the same population (Supplementary Fig. 3) as the evidence integration.

### Despite similar choice-selective sequences, evidence coding differs across regions

Despite similar choice-selective sequences, we found that evidence coding schemes across regions were surprisingly distinct. The neocortical regions primarily had monotonic encoding of evidence, as in the competing chains models, whereas the hippocampus had primarily non-monotonic encoding as in the position-gated bump attractor model. This result emphasizes the importance of examining how variables are represented, not just if they are represented, to compare computational mechanisms and functions across regions[114].

The distinct evidence tuning properties we observed in hippocampus, ACC, RSC, and DMS support current ideas of their function in decision-making. The non-monotonic tuning curves in the hippocampus allow for a direct readout of the evidence based on the location of the active bump. This form of coding is more computationally intensive in the number of neurons required but could provide more refined information about the current state of the animal, supporting a joint cognitive map of position and evidence[48,115,116], as suggested by previous analysis[49]. In contrast, the monotonic encoding of evidence in ACC, RSC, and DMS does not allow a direct readout of evidence from the identity of the active neurons. Instead, it may be ideally suited for making a unitary decision based on the comparison of activity along the two chains, and is similar to the graded coding that has been reported for value or confidence in ACC[117–123] and observed for evidence-encoding in electrophysiological recordings in the accumulating towers task[124]. Interestingly, RSC, while dominated by monotonic tuning curves, did exhibit a population of narrowly tuned neurons. This may reflect that the hippocampus acts as a major input area to RSC, while RSC receives from and projects to other areas of cortex[125–128], providing a possible suggestion as to how information about accumulated evidence is passed and transformed between brain regions. One possibility is that there is a single evidence-accumulation mechanism carried out in one set of brain regions, and other regions read out and transform this evidence into different tuning curve shapes. An alternative possibility is that both evidence accumulation mechanisms are used, but in different brain regions and for different purposes. Finally, the fact that DMS primarily responds in the delay region, where it mainly shows binary choice tuning (Supplementary Fig. 5), aligns with work suggesting the role of the striatum in action

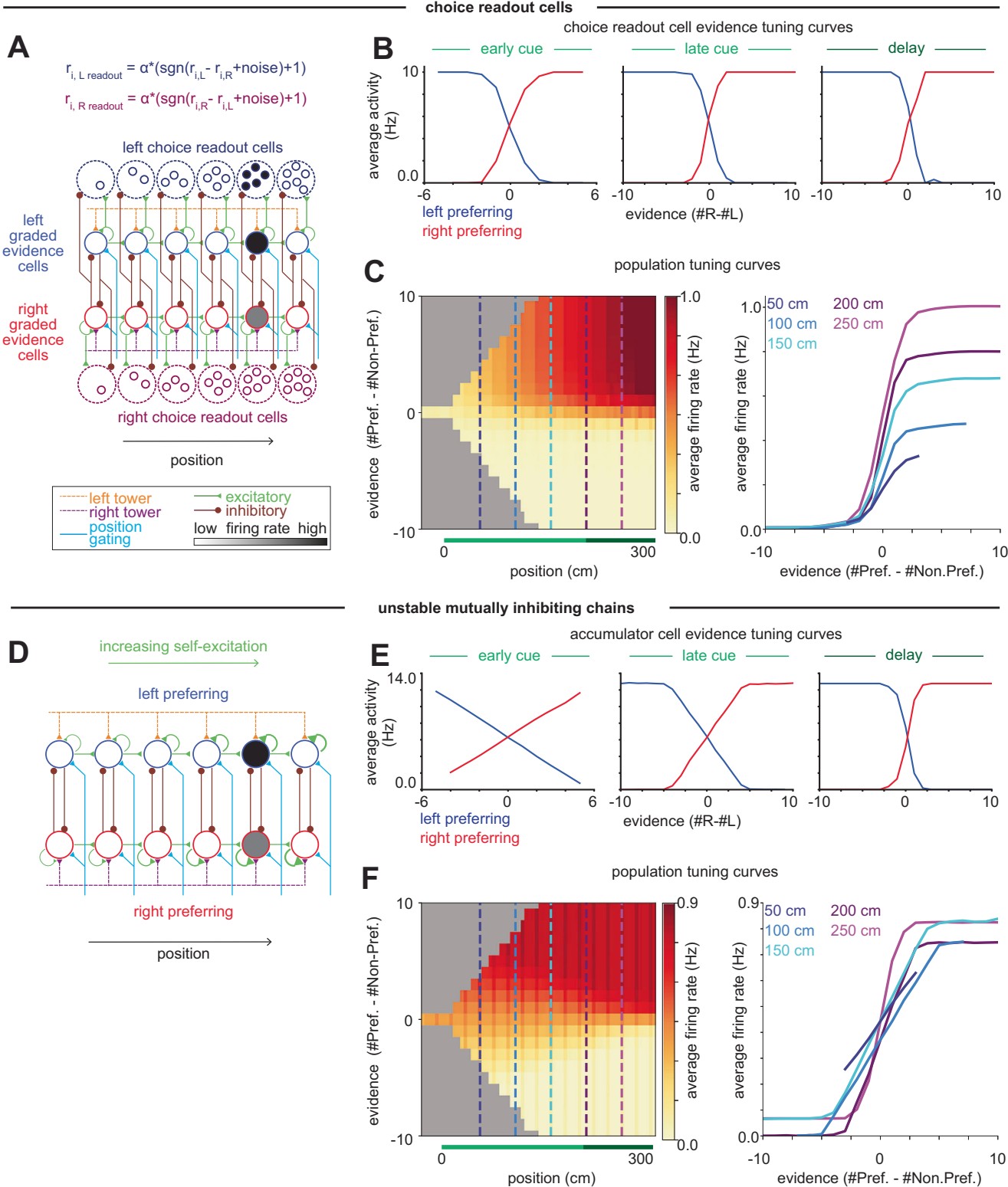

**Fig. 7 | Models that recapitulate the shift across the maze from evidence to choice tuning. A** Model schematic of a population of choice cells that read out from the accumulator and indicate if there is more left evidence (navy blue cells) or right evidence (dark magenta cells). The number of neurons in the choice readout population tuned to each position grows along the chain. **B** Tuning curves of the choice cells at their preferred position. **C** As in Fig. 6L, M, population average tuning curves of the entire population of evidence-tuned cells, including both cells in the accumulator and the choice-readout cells. Cross-sections show a shift towards choice tuning with increasing position in the maze. **D** Schematic of the mutually inhibiting chains model with unstable tuning, in which the strength of self-excitation increases with position along the chain. **E, F** Same as (**B, C**) but for cells in the unstable mutually inhibiting chains model.

selection[129–131], consistent with recordings in a different working memory task[55].

In regions with primarily monotonic evidence tuning (ACC, RSC), we observed that this tuning was graded early in the maze and sharpened to more choice-like tuning later in the cue period and throughout the delay (Fig. 6). While the graded representations of evidence during the early cue period in ACC and RSC are consistent with the competing chains models of evidence accumulation, evidence accumulation models alone do not explain the transition towards more choice-like tuning later in the trial. This transition could reflect a change in dynamics as a result of decision commitment[105,106] or increasing urgency[107,108], and could be produced through many different mechanisms, including through a separate choice readout population or by making the integrator increasingly unstable (Fig. 7)[33]. The observation of a transition across the population from graded evidence coding to binary choice coding is reminiscent of the conclusions from a subpopulation of neurons in another accumulation of evidence task in rats[105] and from the evolution across a trial of the power to predict choice from neural responses in monkeys[40].

We applied our models to interpret neural activity in biological circuits, but the motifs and predictions of these models could also be tested for in trained recurrent neural network models. This could shed light on how less explicitly structured networks perform evidence accumulation[132]. A similar approach has been taken in the study of the path integration circuitry of the entorhinal cortex, where core foundational motifs of connectivity of the style presented here were revealed within the complex circuitry of the trained network[133]. Such studies also may reveal conditions favoring the emergence of evidence accumulation through sequences as opposed to traditional persistent activity, as has been analyzed for other cognitive operations[134]. Thus, tuning curve analysis in both biological and artificial networks may help to determine how these circuits perform the task.

### Functionally targeted optogenetic perturbations could provide causal tests of predicted mechanisms of evidence accumulation

We compared evidence tuning in neurophysiological recording data to those predicted by the models. However, unlike targeted perturbations, correlational analyses do not provide causal evidence for the underlying computations in a region. For example, they cannot distinguish neurons involved in accumulating evidence from those that simply readout evidence accumulated elsewhere. Further, within our set of models, they do not differentiate between the varying competing chains models, whereas optogenetic perturbations can (Fig. 3G and Supplementary Fig. 2F). Recent work, using electron microscopy, has suggested that the synaptic connections in the posterior parietal cortex provide a mutually inhibitory motif for evidence accumulation[135], and these connections could be further probed through direct perturbations. Previous experiments have used optogenetic perturbations to identify brain regions that affect evidence accumulation at different times in the accumulating towers task[136,137], but perturbation experiments that selectively target neurons with specific coding properties[70,138,139] remain to be performed during evidence accumulation. Perturbations in the oculomotor neural integrator, where functionally distinct populations are conveniently anatomically separated, have successfully tested model predictions about the dynamics of the integrator[140–142] and allowed the parameters of a model of this system to be fit directly from neural responses in perturbed and unperturbed trials[141].

Targeted perturbations could provide insight into circuit mechanisms, and more broadly, whether the evidence tuning observed in each region results from local evidence accumulation versus a downstream reflection of evidence accumulated in another region[14,143]. For example, evidence may be encoded in a traditional, non-sequential manner at some location and then combined at a downstream readout with a sequential representation of position to give the observed neural activity. If so, then an experimental prediction would be that activating choice-selective, evidence-encoding neurons that sequentially represent position during the cue region would not increase the firing rates of neurons tuned to later positions nor create a behavioral choice bias. Experiments such as these, combined with the mechanistic models and associated analyses presented in this work, should provide a foundation for elucidating how the neural signals guiding decision-making are accumulated within, and faithfully transferred between, neuronal populations.

## Methods

### The accumulating towers task

The accumulating towers task is a behavioral paradigm, performed in virtual reality, for studying accumulation-of-evidence based perceptual decision making. The complete details of the task and experimental setup can be found in Pinto et al.[59]. Here, we briefly summarize the key features of the task. Head-fixed mice run on an 8-inch styrofoam ball within a virtual environment, generated by ViRMEn[144], projected onto a surrounding screen. The virtual environment has a T-maze structure. In the first 30 cm (the precue region), there are no visual cues. In the next 200 cm (cue region), visual cues appear as towers on both sides, with the total count of towers on each side determined by a Poisson distribution of different means. Towers are positioned randomly with an enforced minimal distance between them. A tower appears when the mouse is within 10 cm of its position and disappears 200 ms later. Following the cue region, there is a 100 cm delay region during which the mouse must remember which side was presented with more towers. At the end of the delay region, the mouse receives a sweetened liquid reward for turning to the side with more cues. For water restriction and shaping protocols, see Pinto et al.[59]. Recording sessions were approximately 1.0–1.5 h in duration, consisting of approximately 150–250 trials per session.

In addition to the accumulation of evidence trials described above, sessions also included some warm-up maze trials (cues on one side only, no distractors, tall visual guides at the end of the maze) at the beginning of the session and interspersed in the session if the animal's performance dropped below some threshold, as described in Pinto et al.[59]. Since these trials do not require evidence accumulation, we did not include them in our analyses.

### Neural recordings

Animal procedures were conducted in accordance with standards from the National Institutes of Health and under the approval of the Princeton University Institutional Animal Care and Use Committee.

The head-fixed setup of the accumulating towers task allows for neural imaging throughout the task. Our neural datasets consist of calcium fluorescence data previously recorded from retrosplenial cortex (RSC, 8 animals, 41 sessions (29 sessions from layers 2/3 and 12 sessions from layer 5), 8579 total neurons)[44] and hippocampus (HPC, 7 animals, 7 sessions, 3144 total neurons)[49], as well as newly acquired recordings from anterior cingulate cortex (ACC, 3 animals, 7 sessions, 1720 total neurons) and dorsomedial striatum (DMS, 8 animals, 11 sessions, 804 total neurons). See below for details of ACC and DMS recordings.

For DMS and ACC data, mice were co-housed with same-sex littermates prior to and following surgical procedures. Mice were maintained on a 12-h/12-h light/dark cycle. All surgical procedures and behavioral training occurred in the dark cycle. Ambient temperature was maintained at 21–26 °C and humidity at 30–70%.

**Animals and surgery for ACC recordings.** Three female mice of CaMKIIa-tTA (JAX 007004) crossed with tetO-GCaMP6s[145] (JAX 024742) were used for two-photon imaging of ACC. These mice, which were heterozygous for each transgene, showed robust and stable expression of GCaMP6s in a subset of CaMKIIa-positive, glutamatergic

neurons in the cortex. All mice were between 12 and 16 weeks when they underwent sterile stereotaxic surgery under isoflurane anesthesia (4% for induction, 1–1.5% for maintenance). The skull was exposed by making a vertical cut with a scalpel. Two craniotomy holes were made, one over ACC (anteroposterior axis: 0.8 mm, mediolateral axis: −0.5 mm, relative to bregma, diameter of ~1.2 mm) and another over DMS (anteroposterior axis: 0.75 mm, mediolateral axis: −1.1 mm, relative to bregma, diameter of ~0.3 mm). Adeno-associated virus (AAV; AAVretro2 serotype) encoding mCherry under CaMKIIa promoter (diluted to $2.5 \times 10^{12}$ genome copies/mL, packaged at Princeton Vector Core) was injected to the DMS (anteroposterior axis: 0.75 mm, mediolateral axis: −1.1 mm, dorsoventral axis: −3.1 and −2.8 mm, relative to bregma) to provide information about a projection target and to be used for better motion correction. A total of 300 nL of AAV was infused at each site along the dorsoventral axis, at a rate of 30 nL/min. After injection, the needle was held in the same place for an additional 10 min. The needle was slowly withdrawn over 10 min to prevent backflow. After injections, the dura below the craniotomy was carefully removed within the ACC craniotomy site, and the cortical tissue above ACC was carefully aspirated up to 300 µm below the surface. Then we implanted a 1.0-mm diameter GRIN lens (4.3 mm length, Inscopix, 1050-002184) into the ACC, down to 1.5 mm in the dorsoventral axis (relative to bregma), using a 3D-printed custom lens holder. After implant, a small amount of diluted Metabond (Parkell) was applied to affix the lens to the skull using a 1-mL syringe and 25-gauge needle. After 10 min, the lens holder grip was loosened while the lens was observed through the stereoscope to make certain that there was no movement of the lens. After applying a more diluted metabond around the lens, a titanium headplate was positioned over the skull using a custom tool and aligned parallel to the stereotaxic frame using an angle meter. The headplate was affixed to the skull using a Metabond. Finally, a titanium ring to hold the immersion medium was placed on the headplate, with the lens in the middle, and then affixed to the headplate using dental cement blackened with carbon. After the dental cement was fully cured, mice were unmounted from the stereotaxic frame and protective head caps were placed. Recovery of the mice was monitored for ~2 h. After 1 or 2 weeks of recovery, mice started water-restriction, and after 5 days, behavioral training for the accumulating towers task began, as described in Pinto et al.[59]. Histology showed that GRIN lenses sampled mostly deep layers (layers 5 and 6) in two animals, while in one mouse a GRIN lens was medially implanted to cover both superficial (layers 2 and 3) and deep layers. For the data analysis, we did not make distinctions between neurons in different layers but instead pooled neurons from all layers together.

**Animals and surgery for DMS recordings.** D1R-Cre ($n = 3$, female, EY262Gsat, MMRRC-UCD), A2a-Cre ($n = 3$, male, KG139Gsat, MMRRC-UCD), and D2R-Cre ($n = 2$, male, ER44Gsat, MMRRC-UCD) mice were backcrossed to a C57BL/6J background (Jackson Laboratory, 000664) and bred in-house. All transgenic mice were heterozygous for the transgene. These mice were used for DMS two-photon imaging. Surgical procedures were identical to those described above for the ACC except for the following details. Approximately 1 week prior to GRIN lens implantation, a Cre-dependent AAV expressing GCaMP6f under either the CAG promoter (AAV5-CAG-DIO-GCaMP6f, Addgene, diluted to 3.5 ×1012 genome copies/mL) or synapsin promoter (AAV1-Syn-DIO-GCaMP6f, diluted to 3.8 ×1012 genome copies/mL) was delivered (500 ul per injection site; 200 ul/min) to the DMS (anteroposterior axis: 0.7 mm, mediolateral axis: 1.15 and 1.85 mm, dorsoventral axis: −3 mm, relative to bregma) and the surgical incision was closed using monofilament nonabsorbable sutures (Johnson and Johnson). This allowed for expression of GCaMP6f in D1R- or D2R-expressing medium spiny neurons according to the transgenic mouse line used, and facilitated the settling of brain tissue prior to GRIN lens implantation. Following recovery (5–10 days), sutures were removed, and the viral

injection craniotomy was expanded and dura removed to accommodate a 1.0-mm diameter GRIN lens with a prism (4.3 mm length, Inscopix, 1050-004601). The lens was implanted with prism window facing posterior and just anterior to viral injection sites (anteroposterior axis: 0.75 mm, mediolateral axis: 1.35 mm, dorsoventral axis: −3.25 mm, relative to bregma). Fixation of the lens to the skull, as well as the custom-printed headplate, water immersion ring, and protective covering were identical to that described above for ACC. Across all DMS targeted GCaMP6f imaging mice we obtained three distinct fields of view from one mouse, two distinct fields of view from a second mouse, and the remaining mice contributed single fields of view (8 animals, 11 sessions, 804 total neurons).

**Optical imaging and data acquisition for ACC and DMS recordings.** Imaging was performed using a custom-built two-photon microscope compatible with virtual reality (VR). The microscope was equipped with two high-power femtosecond lasers (Alcor, Spark Laser), with fixed wavelengths for 920 and 1064 nm. The scanning units used a 5-mm galvanometer and a 8 kHz resonant scanning mirror (Novanta). The collected photons were passed to a dichroic mirror (XYZ) that separates near-infrared light from the visible light and then split into two channels by another dichroic mirror (FF562-Di03, Semrock). The light for respective green and red channels was filtered using a band-pass filter (FF01-520/60 and FF02-607/70, Semrock), and then detected using GaAsP photomultiplier tubes (1077PA-40, Hamamatsu). The signal from the photomultiplier tubes was amplified using a high-speed current amplifier (59–179, Edmund). Output of the amplifier was sent to a data acquisition system (National Instruments). Black rubber tubing was attached around the objective (Zeiss, W N-Achroplan 20x/0.5 M27, ×20 magnification, 0.5 NA) to shield the light from the virtual reality screen. The objective was aligned to the implanted GRIN lens within the implanted titanium headring. Double-distilled water was filled inside the headring and used as the immersion medium. The amount of laser power at the objective was 30–40 mW for ACC and 40–50 mW for DMS. Control of the microscope and imaging acquisition was performed with the ScanImage software (MBF Bioscience), on a separate computer from the VR-running one. Images were acquired at 30 Hz with a size of $512 \times 512$ pixels. Synchronization between behavioral logs and acquired images was achieved by sending behavioral information from the VR controlling computer to the scanning computer via an I2C serial bus each time the ViRMEn environment was refreshed. Specifically, numeric information about the current session, trial, and frame were written in the header of TIFF imaging files.

**Data processing for ACC and DMS recordings.** Both ACC and DMS imaging data were processed with Suite2p. For ACC recordings in which mCherry signals are present, rigid motion correction was performed with that channel and then applied to the functional green channel with GCaMP6s, with the following parameter settings: nimg_init = 1000, smooth_sigma = 1.15, maxregshift = 0.1. For DMS recordings, the same motion correction was performed, but with the green channel. Regions-of-interest (ROIs) corresponding to individual neurons were detected, and calcium signals were extracted and neuropil-corrected (neucoeff = 0.7). Subsequently, an experimenter-mediated curation was performed on these ROIs to discriminate putative neurons from noise. Neuropil-corrected fluorescence from these curated neurons was exported to MATLAB, and the $\Delta F/F$ was calculated, taking baseline fluorescence as the mode of the fluorescence within 3 min long windows (the same baseline as applied in the RSC and HPC datasets).

**Spike inference.** In order to compare neural recordings across datasets with different calcium indicators, we performed spike inference on our data based on a recently developed and validated algorithm that has only two free parameters, the average estimated firing rate

and the calcium decay rate[146,147]. For each $\Delta F/F$ dataset, we assumed a baseline firing rate of 2 Hz (but our results were not sensitively dependent on this specific choice) and set the calcium decay rate according to the values fit by Jewell et al.[146] for each calcium indicator (0.9885 for GCaMP6s (ACC), 0.9704588 for GCaMP6f (all other regions)). From these inferred spikes, we inferred a continuous firing rate by smoothing with a Gaussian window function (window length = 1 s, standard deviation = 0.25 s).

**Position binning of neural and evidence data.** To compare neural data along maze positions and evidence levels, we resampled the neural (continuous firing rate as calculated above) and behavioral (cumulative evidence up to that point) data in position bins. Specifically, firing rates were averaged based on the position of the animal within 5 cm bins from the start of the trial (−30 cm) to the end of the delay region (300 cm) for each trial. Cumulative evidence for each bin was defined by the number of right minus the number of left cues observed before the start of the position bin.

**Trial selection criteria.** We discarded a small fraction of trials from analysis using the following criteria. As mentioned above, any trials from the warm-up maze were excluded, as these trials do not require evidence accumulation. Additionally, trials with excessive travel were discarded (3.2% of trials in ACC sessions, 8.0% of trials in DMS sessions), which were identified as trials where mice exhibited a large view angle deviation (more than 90 degrees, either left or right) before the end of the delay region.

### Determining choice selectivity of neurons

To identify choice-selective neurons (Fig. 1B–E), we first jointly fit the position and evidence tuning of each neuron in our dataset as described in Data Analysis below. We then considered only neurons that had significant evidence tuning. To produce the sequence plots, neurons were first sorted as right-preferring if the mean of the neuron's evidence tuning was greater than 0 and left-preferring otherwise and then sorted by their position mean. Using this sorting, we then took the average activity of each neuron at different positions for correct left and correct right trials. The resulting traces for each neuron were then normalized to the peak level of average activity.

### Traditional integrator models simulations

To show the activity of traditional integrator models (Fig. 2), we simulated equations corresponding to previously developed models of integration: a competing accumulator model and a bump attractor model.

**Competing accumulator model.** We modeled two competing populations of neurons, one left-preferring population and one right-preferring population. The activity $r_L$ of the neuron representing the left population is given by

$$\frac{dr_L}{dt} = -ar_L + [br_L - er_R + f1_{\text{left}}(t) + X(t)]^+$$

and a corresponding equation governs the right population, where

$$1_{\text{left}}(t) = \begin{cases} 1 & \min_{\ell \in L} |p(t) - \ell| < 0.5\text{cm} \\ 0 & \text{otherwise} \end{cases}$$

and

$$1_{\text{right}}(t) = \begin{cases} 1 & \min_{R \in \mathcal{R}} |p(t) - R| < 0.5\text{cm} \\ 0 & \text{otherwise} \end{cases}$$

where $p(t)$ is the position of the animal at time $t$, $L$ is the set of positions of left cues, and $\mathcal{R}$ is the set of positions of right cues.

In our simulations, we used parameters $a = 50\,\text{s}^{-1}$, $b = 10\,\text{s}^{-1}$, $e = 40\,\text{s}^{-1}$, $f = 100$ Hz/s, and a common external input to both populations, $X(t) = 500$ Hz/s. The model was simulated as described in the simulation of differential equations.

**Bump attractor model.** For the bump attractor model, we assumed each neuron has its peak response at a different evidence level. The activity of the neuron with peak response at evidence $j$ is given by

$$\frac{dr_j}{dt} = -ar_j + F\left(\sum_k W_{j,k}r_k + c\left(I_{j+1,L}(t) + I_{j-1,R}(t)\right) + X(t)\right),$$

where

$$F(x) = \frac{q(1 + \tanh(\gamma x))}{2},$$

$$I_{j,L}(t) = r_j 1_{\text{left}}(t),$$

and

$$I_{j,R}(t) = r_j 1_{\text{right}}(t)$$

where

$$1_{\text{left}}(t) = \begin{cases} 1 & \min_{\ell \in L} |p(t) - \ell| < 0.5\text{cm} \\ 0 & \text{otherwise} \end{cases}$$

and

$$1_{\text{right}}(t) = \begin{cases} 1 & \min_{R \in \mathcal{R}} |p(t) - R| < 0.5\text{cm} \\ 0 & \text{otherwise} \end{cases}$$

where $p(t)$ is the position of the animal at time $t$, $L$ is the set of positions of left cues, and $\mathcal{R}$ is the set of positions of right cues.

We assigned the neuron at each evidence level an angle, evenly spaced between 0 and $2\pi$, and set the synaptic connection weights as

$$W_{j,k} = \omega_0\left(\cos\left(\theta_j - \theta_k\right) + \omega_1\right),$$

We note that, since evidence is not periodic, a sufficiently large number of evidence levels were used such that substantial interactions do not occur between the nominally periodically connected endpoints.

For the shown simulation, we used 35 neurons, corresponding to evidence levels of −17 to 17. We used the parameters $a = 55\,\text{s}^{-1}$, $X(t) = 0.04$ Hz/s, $q = 1250$ Hz/s, $\gamma = 1$ s/Hz, $\omega_O = 0.12\,\text{s}^{-1}$, $\omega_1 = -1.0$, and $c = 0.2\,\text{s}^{-1}$. Neurons $r_{-1}$, $r_0$, and $r_1$ were initialized to 15, 17.5, and 15 Hz activity, respectively, and all other neurons were initialized to 0 Hz activity. The model was simulated as described in the simulation of differential equations.

### Parameterization of the mutually inhibiting chains model

As in Eq. 1, the activity of the neuron at position $i$ in the left chain can be described as

$$\frac{dr_{i,L}}{dt} = -ar_{i,L} + \left[br_{i,L} + cr_{i-1,L} - er_{i,R} + f1_{\text{left}}(t) + P_i(t) - T\right]^+,$$

and a corresponding equation governs the activity of neurons in the right chain.

A neuron's active position is defined by a square pulse over a range of positions, where $p(t)$ is the position of the animal at time $t$,

$$P_i(t) = \begin{cases} T + X(t) & i \le \frac{p(t)}{P_0} < (i+1) \\ 0 & \text{otherwise} \end{cases}$$

The external input is of the form

$$1_{\text{left}}(t) = \begin{cases} 1 & \min_{\ell \in L} |p(t) - \ell| < 0.5\,\text{cm} \\ 0 & \text{otherwise} \end{cases}$$

and

$$1_{\text{right}}(t) = \begin{cases} 1 & \min_{R \in \mathcal{R}} |p(t) - R| < 0.5\,\text{cm} \\ 0 & \text{otherwise} \end{cases}$$

where $L$ is the set of positions of left cues, and $\mathcal{R}$ is the set of positions of right cues.

In our simulations, we used parameters $a = 50\,\text{s}^{-1}$, $b = 10\,\text{s}^{-1}$, $c = 50\,\text{s}^{-1}$, $e = 40\,\text{s}^{-1}$, $f = 100\,\text{Hz/s}$, $T = 15000\,\text{Hz/s}$, and $X(t) = 500\,\text{Hz/s}$. (See Supplementary Text for necessary conditions on the parameters.) We take $P_0 = 20\,\text{cm}$, which divides the total length of the maze (330 cm) into 17 different sections of active positions. Therefore, in these models, we simulate 17 neurons in each chain for a total of 34 neurons. This choice of parameters was selected to recapitulate typical firing rates of the neurons that we recorded, as well as approximate the typical position width observed in the experimental choice-selective sequence plots (Fig. 1).

The parameters above produce saturation in the evidence tuning at the extreme evidence levels (Fig. 3E). As demonstrated in Supplementary Fig. 2J, for different parameters, this model will instead show linear tuning curves over the observed range of evidence (same parameters as above, but $X(t) = 1000\,\text{Hz/s}$). We note that a similar effect of linear evidence tuning throughout the observed evidence range could have been achieved by decreasing the weight $f$ on the cues. Both manipulations prevent the non-dominant chain from reaching 0 Hz, which results in saturation of the dominant chain.

In Supplementary Fig. 2G, we used the original parameters but took $T = 1100\,\text{Hz/s}$, and we changed the form of the position gate to be a smooth Gaussian, rather than a square pulse,

$$P_i(t) = \alpha \exp\left(\frac{-(p(t) - \mu_i)^2}{\beta}\right)$$

where $\alpha = 60\,\text{Hz/s}$, $\beta = 1250\,\text{cm}^2$, and $\mu_i = 20i - 10\,\text{cm}$.

In Supplementary Fig. 2H, we change the form of the position gate to have heterogeneous widths. We use the original parameters but only 12 neurons in each chain, and rather than a fixed width $P_0$ for each position, we draw a set of transition points $\rho_i$ uniformly between $-30$ and 300 cm. Then, to avoid overly close together transition points, if $(\rho_{i+1} - \rho_i) < 5\,\text{cm}$, we increment $\rho_{i+1}$ by 5 cm. We then define

$$P_i(t) = \begin{cases} T + X(t) & \rho_i \le p(t) < \rho_{i+1} \\ 0 & \text{otherwise} \end{cases}$$

### The mutually inhibiting chains model for the transition to choice representations

In Fig. 7, we propose two models that could explain the transition from more graded evidence representations early in the maze to more choice-like representations later in the maze. The first model (Fig. 7A–C) incorporates a population of choice-readout cells. In this model, the evidence accumulator cells of the chains are modeled exactly as described in the previous section, but we introduce a population of choice readout cells at each position. The firing rate of a left choice readout cell at position $i$, $r_{i,\,L,\text{readout}}$, is given by

$$r_{i,L,\text{readout}} = \alpha\left(\text{sgn}\left(r_{i,L} - r_{i,R} + \xi\right) + 1\right)$$

where $\alpha = 5\,\text{Hz}$, sgn is the signum function which returns the sign of its input, and $\xi \sim \mathcal{N}(0, 2)$ Hz is a Gaussian noise term, with the analogous equation for a right choice readout cell. We assume that there are $i$ choice readout cells for each choice at position $i$.

In the second model (Fig. 7D–F), we set the parameters of the mutually inhibiting chains model such that the integrator becomes increasingly more unstable with advancing position down the chain. Specifically, we incrementally increase the weight of self-excitation $b$ in Eq. 1 with progression down the chain by making it a function of the position $i$:

$$b_i = b_0 + \Delta i$$

where $\Delta = 0.5\,\text{s}^{-1}$, and $b_0 = 10\,\text{s}^{-1}$ is the value of self-excitation that gives a perfect integrator.

### Parameterization of the position-gated bump attractor

In Eq. 2, we defined the evolution of the firing rate for a neuron at position $i$ and evidence $j$ as

$$\frac{dr_{i,j}}{dt} = -ar_{i,j} + F\left(\sum_k W_{j,k} r_{i,k} + br_{i-1,j} + c\left(I_{i,j+1,L}(t) + I_{i,j-1,R}(t)\right) + P_i(t) - T\right)$$

A neuron's active position is defined by a square pulse over a range of positions, where $p(t)$ is the position of the animal at time $t$,

$$P_i(t) = \begin{cases} T + X(t) & i \le \frac{p(t)}{P_0} < (i+1) \\ 0 & \text{otherwise} \end{cases}.$$

The neuronal nonlinearity is

$$F(x) = \frac{q(1 + \tanh(\gamma x))}{2}.$$

The strengths of the synaptic connections between neurons were chosen to be symmetric with excitatory connections between neurons nearby in their evidence tuning and inhibitory connections between neurons farther apart in their evidence tuning. This was done by assigning each evidence level an angle, evenly spaced between 0 and $2\pi$, and setting

$$W_{j,k} = \omega_0\left(\cos\left(\theta_j - \theta_k\right) + \omega_1\right).$$

We note that, since evidence is not periodic, a sufficiently large number of evidence levels were used such that substantial interactions do not occur between the nominally periodically connected endpoints.

When an input is present, asymmetric connections cause the active evidence neurons to shift towards the input. This is modeled through shifter neurons (Fig. 4A, purple and orange small circles) that are activated by the coincidence of external cue signals and neural activity from the current location of the bump by

$$I_{i,j,L}(t) = r_{i,j} 1_{\text{left}}(t),$$

and

$$I_{i,j,R}(t) = r_{i,j} 1_{\text{right}}(t)$$

where

$$1_{\text{left}}(t) = \begin{cases} 1 & \min_{\ell \in L} |p(t) - \ell| < 0.5\,\text{cm} \\ 0 & \text{otherwise} \end{cases}$$

and

$$1_{\text{right}}(t) = \begin{cases} 1 & \min_{R \in \mathcal{R}} |p(t) - R| < 0.5\,\text{cm} \\ 0 & \text{otherwise} \end{cases}$$

where $p(t)$ is the position of the animal at time $t$, $L$ is the set of positions of left cues and $\mathcal{R}$ is the set of positions of right cues.

As in the competing chains models, the position-gating of each layer of neurons is activated over a length 20 cm ($P_0 = 20$ cm), again giving 17 different active positions. For the neurons to span the range of evidence of the experiments, we used 35 neurons in the evidence dimension, corresponding to evidence levels of −17 to 17. Thus, we simulated a total of 595 neurons.

The parameters used in the simulation are: $a = 55\,\text{s}^{-1}$, $b = 0.088\,\text{s}^{-1}$, $T = 300\,\text{Hz/s}$, $X(t) = 0.04\,\text{Hz/s}$, $q = 1250\,\text{Hz/s}$, $\gamma = 1\,\text{s/Hz}$, $\omega_O = 0.12\,\text{s}^{-1}$, $\omega_1 = -1.0$, and $c = 0.2\,\text{s}^{-1}$. Neurons $r_{0,-1}$, $r_{0,0}$, and $r_{0,1}$ are initialized to 15, 17.5, and 15 Hz activity, respectively, and all other neurons are initialized to 0 Hz activity.

### Simulation of differential equations

Differential equations for each model (Eqs. 1, 2, and S2) were simulated in Python using scipy.odeint, which uses the lsoda algorithm, with a maximum stepsize of 0.01 s (or equivalently 0.5 cm when assuming a constant velocity of 50 cm/s) to find the activity levels of the neurons in each model for points between −30 and 300 cm in 0.1 cm increments. In all simulations, we assume the animal travels at a constant velocity of 50 cm/s through the maze so that time and position are interchangeable. We simulated the neural activity for 1000 trials with cue positions derived from 1000 experimental trials from the behavioral and neural recording data. For the competing chains models, we say that the animal makes a left choice if the activity of the last neuron in the left chain is greater than the activity of the last neuron in the right chain. For the bump model, we say that the animal makes a left choice if the index $k$ of the neuron with maximal firing in the last layer is less than $0.5J$, where $J$ is the total number of neurons in the layer. If the maximal firing occurs exactly at $0.5J$, we determine the animal's decision by randomly choosing between the two alternatives with equal probability.

**Simulations with input noise.** For each model, we simulated 25 sessions with input noise (Figs. 3C and 4C and Supplementary Fig. 2B). For each session, we selected 150 trials drawn randomly without replacement from the 1000 experimental trials that we used for simulation of the model. Let $L$ and $R$ be the set of left cues and right cues, respectively, in the original trial. For each cue $x$ in $L$, we drew $z$, a random uniform number between 0 and 1, and if $z < 0.33$, we added $x$ to the set $L'$. We repeated the same process for each cue in $R$, yielding two reduced sets of cues when input noise is present, $L'$ and $R'$, corresponding to the animal ignoring or not perceiving 67% of the cues. We then simulated the model with the input terms $1_{\text{left}}(t)$ and $1_{\text{right}}(t)$ determined according to the reduced cue sets. We then calculated the psychometric curves (as described in "Psychometric curves" below) with the difference in cues based on the original cue sets, but whether the animal turns left or right based on the simulation of the trial with the reduced cue sets.

**Evidence tuning curves for simulated model data.** In order to find the evidence tuning curves (Figs. 3E and 4E and Supplementary Fig. 2D, J, K), we averaged the activity of each neuron binned by position and accumulated evidence for each of the 1000 simulated trials. Position

bins were of size 0.1 cm, ranging from −30 to 300 cm. Evidence bins were of size 1 tower, ranging from −15 to 15 towers. Bins that had fewer than 2 samples were not included in further analysis and not plotted. For a given neuron, we averaged across evidence levels to find the position with maximum firing. We then plotted the average firing rate versus evidence level at this position.

**Simulated single-neuron perturbations.** Single-neuron perturbations were simulated in the same manner as no-perturbation trials, but in the absence of any cues. A subthreshold excitatory term $O_i(t)$ was added within the thresholded dynamics of each model (i.e. within the square brackets of Eq. 1 or within the function $F$ of Eq. 2) to simulate the effects of stimulating an individual neuron, where $O_i(t) = 25\,\text{Hz/s}$ for the mutually inhibiting competing chains model, $O_i(t) = 12.5\,\text{Hz/s}$ for the uncoupled competing changes model, and $O_i(t) = 0.2\,\text{Hz/s}$ for the position-gated bump attractor, if neuron $i$ is the neuron being stimulated, and 0 otherwise.

### Psychometric curves

For each session, we sorted trials based on the difference in the final number of cues at the end of the maze. We binned these trial differences with bin size of 3 towers, ranging from −14 to 14 towers. Within each bin, we then found the fraction of trials in each bin for which a left choice was made (Figs. 3C and 4C and Supplementary Fig. 2B).

### Data analysis

**Determining position and evidence selectivity.** In our neural recording data, we sought to identify the set of neurons whose firing encoded accumulated evidence. As described in the main text, due to the nonuniform sampling of position and evidence and the fact that neurons are only active at some positions, we found that robustly identifying evidence selectivity required that we jointly identify the position range over which a neuron is active and the evidence selectivity within this set of positions. To do so, we assumed the firing rate of each neuron had the form,

$$FR(p, e) = aP(p)E(e) + b$$

where $FR(p,e)$ is the firing rate of the neuron at position $p$ and evidence $e$, and $P$ and $E$ were described by Gaussians,

$$P(p) = \exp\left(\frac{-\left(p - \mu_p\right)^2}{2\sigma_p^2}\right)$$

and

$$E(e) = \exp\left(\frac{-(e - \mu_e)^2}{2\sigma_e^2}\right).$$

The above functional form, which assumes evidence and position tuning can be approximated by independent Gaussians, was not used as a detailed descriptor of neuronal firing (for less constrained fits around a given position, see "Neural evidence tuning curves around the most active position"). Rather, we found that it was effective for the purpose of identifying evidence-modulated neurons and determining if the mean evidence selectivity favors left or right evidence (as used to sort neurons by evidence-selectivity in Fig. 1B–E). Here, the Gaussian was allowed to extend beyond the range of observed evidence values, so that it can accommodate both monotonic and non-monotonic tuning curves (i.e., if the mean evidence occurs at the extremes of the evidence range, the evidence tuning will be monotonic throughout the range, Supplementary Fig. 4D).

For each neuron, we normalized the firing rate observations (binned as in "Position binning of neural and evidence data") to the

range [0, 1]. We then fit the parameters of the above model using an iterative fitting procedure, alternating fitting $\mu_p$, $\sigma_p$, $a$, and $b$ with fitting $\mu_e$, $\sigma_e$, $a$, and $b$ with a nonlinear fitting method (scipy curvefit, which employs the trust region reflective algorithm, with bounds $-50 \leq \mu_p \leq 350$, $0 \leq \sigma_p \leq 200$, $e_{low}-1 \leq \mu_e \leq e_{high}+1$, $0 \leq \sigma_e \leq 30$, $0 \leq a \leq 10$, and $0 \leq b \leq 1$). The bounds on $\mu_e$ are defined by $e_{low} = \min(E')$ and $e_{high} = \max(E')$, where $E'$ is the set of all evidence levels observed by the neuron within 5 cm of $\mu_p$. These bounds are updated on each iteration based on the current fit of $\mu_p$. Due to the iterative nature of our fitting procedure, after the final position update, $\mu_e$ may no longer satisfy these bounds, and we do not include such neurons when analyzing the distributions of the normalized evidence parameters.

We then tested for significance of the evidence tuning fit using the pseudosession method[148]. For each session of firing rate and position data, we generated new evidence for each trial in the session by resampling trials from actual sessions, and applied the iterative fitting procedure to the new pseudosession evidence and position data. We repeated this process for 50 pseudosessions and calculated the mean squared error between the true firing rates and the model predictions for each session. We then used a one-sample t-test to compare the fit of the model, as measured by the mean squared error, in the true session to the fit of the null distribution from the pseudosession, and only considered neurons for which the fit was significantly better ($p < 0.05$) than the null distribution for further analysis.

In order to compare the parameterization of the evidence tuning across positions (Fig. 5D, H, L and Supplementary Fig. 5D), we normalized $\mu_e$ and $\sigma_e$ to account for the different observed evidence ranges at different positions. Let $\mathcal{E}$ be the set of all evidence values observed by the neuron at positions within 5 cm (one position bin) of $\mu_p$. Define $\mathcal{E}^+ = \{e \in \mathcal{E} | e > 0\}$, i.e., the set of positive observed evidence levels, and $\mathcal{E}^- = \{e \in \mathcal{E} | e < 0\}$. For $\mu_e < 0$, the normalized mean is given by

$$\overline{\mu_e} = \frac{\mu_e}{\max_{e \in \mathcal{E}}(|e|)},$$

and for $\mu_e > 0$, the normalized mean is given by

$$\overline{\mu_e} = \frac{\mu_e}{\max_{e \in \mathcal{E}^+}(e)}$$

The normalized evidence standard deviation is given by

$$\overline{\sigma_e} = \frac{\sigma_e}{\max(\mathcal{E}) - \min(\mathcal{E})}$$

For visualization purposes (Fig. 5 and Supplementary Fig. 5), we plot only the 80% of significantly evidence-selective neurons in each region with the best fit. For robustness to single large observations making a neuron appear narrowly tuned, for neurons with $\sigma_e < 3$, we identified the nearest position-evidence bin to $(\mu_p, \mu_e)$. If there were fewer than 4 observations in this bin or if there was a single outlier in this bin (defined by one observation greater than 1.5 times the interquartile range), the neuron was excluded from plotting. For visualization of all significantly evidence-selective neurons without any further criteria applied, see Supplementary Fig. 12. In all density plots, kernel density estimates were generated using the Python package seaborn, which uses scipy.stats.gaussian_kde to estimate the density with Gaussian kernels (bandwidth determined by Scott's Rule).

**Neural evidence tuning curves around the most active position.** To generate the 1-D evidence tuning curves around the most active position (Fig. 5B, F, J and Supplementary Fig. 5B) for each neuron with significant evidence tuning, we identified the subset of data within $0.5\sigma_p$ of $\mu_p$. (Note that if this range extended to less than 0 cm or greater than 300 cm, only the data within 0–300 cm were used.) Using the data from within this range, the tuning curves were computed by

computing the average of the firing rates at each evidence level. We then fit the averages to two functional forms, a Gaussian and a logistic function, and plotted the curve that better fit the average tuning curve. By fitting to the averages, rather than the individual data points, we found we could better account for the uneven distribution of sampling across evidence levels at different positions (Supplementary Fig. 4A). The Gaussian evidence tuning curve fit took the form

$$FR(e) = a \exp\left(\frac{-(e - \mu)^2}{\sigma^2}\right) + c$$

where $FR(e)$ denotes the firing rate as a function of the evidence $e$. The fit was performed with a nonlinear fitting method (scipy curvefit, which employs the trust region reflective algorithm, with bounds $m < \mu < M$, $0 < \sigma < 30$, $0 < a < 10$, $0 < c < 1$, where $m$ is the minimum and $M$ is the maximum observed evidence within the active position). The logistic evidence tuning curve took the form

$$FR(e) = \frac{a}{(1 + \exp(-k(e - x_0)))} + c$$

and was fit with the same nonlinear fitting method (scipy curvefit, with bounds $-1 < k < 1$, $-15 < x_0 < 15$, $0 < a < 10$, $0 < c < 1$). To determine the curve with the better fit, we calculated the mean squared error between the predicted firing rates from each fit curve and the true firing rates. We note that each form of the tuning curve has the same number of fit parameters, so there was no need to correct for the numbers of parameters when comparing model predictions.

For comparison to our fit tuning curves, we also plotted the raw evidence tuning curves around the most active position (Supplementary Fig. 6). We note that such raw evidence tuning curves around a given position require a reasonable assessment of the peak position tuning, as was obtained in our joint position-evidence fits described above. However, once the most active positions were obtained, the fits no longer depended on requiring the evidence to have the same functional form at every position or, in the case of the raw fits, on a particular functional form. Thus, we found that the fits could provide a more quantitatively precise visualization of the evidence tuning around the most active position and, for the neurons best fit by logistic functions, a better quantitative assessment of the steepness of the sigmoid. However, beyond this small quantitative difference, the fits typically did not differ substantially from the joint Gaussian fits, and required fitting to two separate functional forms, so we used the joint Gaussian fits for the population-wide assessment of tuning in Fig. 5C, D, G, H, K, L.

**Cue encoding model.** To find the average response to a cue (Fig. 6A–E), we fit a linear cue encoding model to the difference between the average right-preferring activity and average left-preferring activity on each session, using the subset of neurons that are significantly evidence tuned:

$$f_{pop}(p) = \beta_0 + \sum_{\ell \in L}(K_L * \ell)(p) + \sum_{r \in R}(K_R * r)(p)$$

where $f_{pop}(p)$ is the difference between the average activity of the right evidence-preferring cells and the average activity of the left evidence-preferring cells at position $p$, $L$ is the set of left cues, $R$ is the set of right cues, and $K_L$ and $K_R$ are the cue response kernels to left and right cues respectively. As above, neurons that were significantly evidence tuned were categorized as right preferring if they had $\mu_e > 0$ and left preferring if $\mu_e < 0$.

For this analysis, we filtered the inferred spikes from the $Ca^{++}$ imaging data using a causal half-Gaussian (window length = 1 s, standard deviation = 0.25 s), so that responses to cues would be purely

causal. In order to fit the shapes of the cue response kernels of width 300 cm, each cue was convolved with a cubic spline basis set with 7 degrees of freedom[149]. We fit a ridge regression model with regularization strength 1 to predict the difference in population activity. The regression used the convolution of the cues with each of these splines as regressors, so that the cue encoding model was given by

$$f_{pop}(p) = \beta_0 + \sum_{k}^{N_s} \beta_{k,L}(s_k * \ell)(p) + \sum_{k}^{N_s} \beta_{k,R}(s_k * r)(p)$$

where $s_k$ is the $k$th basis function, $N_s = 7$ is the number of basis functions, and $\ell$ and $r$ are binary vectors with value 1 at the onset (defined as the time at which the cue becomes visible, 10 cm ahead of its position) of each left and right cue respectively.

Left cues and right cues were convolved separately to derive separate cue response kernels for each cue type. In Supplementary Fig. 13A, B, separate cue kernels were fit for left and right cues when the magnitude of the current evidence prior to cue appearance, $e$, was low ($|e| \leq 1$), medium ($2 \leq |e| \leq 4$), or high ($|e| \geq 5$). For the medium and high evidence cues, the width of the cue kernel was 200 cm. In Supplementary Fig. 13C, D, separate cue kernels were fit for left and right cues when the cue occurred in the early (before 70 cm), middle (between 70 and 140 cm), or late (after 140 cm) part of the cue period. For early cues, the width of the cue kernel was 300 cm, for middle cues, the width of the cue kernel was 230 cm, and for late cues, the width of the cue kernel was 160 cm.

**Single-neuron evidence vs. choice encoding analysis.** For each neuron at each position bin $p$, we fit a single-neuron evidence vs. choice encoding model (Fig. 6F–H) to describe the firing rate of the $i$th neuron

$$FR_i(e,c,p) = \beta_0 + \beta_1 e(p) + \beta_2 c$$

as a function of the current evidence $e(p)$ at that position and the upcoming behavioral choice on that trial ($c$), using ordinary least squares regression. For this analysis, we fit to data from both correct and incorrect trials. For each coefficient, we calculated the F-statistic, and used the F-test (one-sided) to determine whether a coefficient was significant at a 0.01 significance level. At each position, for each session, we found the fraction of active neurons at that position that had a significant coefficient. For this analysis, a neuron was defined as active at position $p$ if $|p - \mu_p| < \sigma_p$, based on the results of the joint position-evidence fitting procedure described above.

**Average evidence population responses for trials with different evidence levels.** In Supplementary Fig. 13E, F, we examined the average population responses across positions for correct trials with different final evidence levels. We binned trials by final evidence for left and right trials for low ($1 \leq e \leq 3$), medium ($4 \leq e \leq 6$), and high evidence levels ($e \geq 7$). At each position bin, we only analyzed neurons that met two criteria: (1) they were active at this position bin, where active at position $p$ is defined as above as having $|p - \mu_p| < \sigma_p$, and (2) had a significant evidence coefficient $\beta_1$ (at a 0.01 significance level) at any position bin, based on the results of the single-neuron encoding model described above. For these neurons, we calculated the difference between the average firing rates (extracted from the Ca$^{++}$ imaging using the causal half Gaussian filter, as described above) of the right-preferring and left-preferring populations. As above, a neuron is defined as right preferring if $\mu_e > 0$ and left preferring if $\mu_e < 0$.

**Linear evidence decoding model.** To evaluate the ability to linearly decode evidence from the population on correct trials (Fig. 6I–K), we fit two decoders at each position bin, one for trials in which the evidence was greater than or equal to 0 and one for trials for which the

evidence was less than or equal to zero. Separate decoders were calculated so that positive correlations between actual and decoded evidence do not result just from correctly decoding the animal's choice on correct trials. To further eliminate correlations resulting from predicting just the sign-of-evidence, we also applied the same analysis but with one decoder for evidence strictly greater than zero and one for evidence strictly less than zero (Supplementary Fig. 13G, H).

For each decoder, we used nested five-fold cross-validation[150] to fit a ridge regression model to predict evidence from the activity of the active neurons at that position. As above, neurons were defined as active at position $p$ if $|p - \mu_p| < \sigma_p$. In the inner loop of cross-validation, the regularization strength was chosen from [0.0001, 0.001, 0.01, 0.1, 1, 10, 100, 1000], with the best performing regularization strength used for evaluation on the test set in the outer loop. Decoders were evaluated by the average correlation between the predicted and actual evidence in the test set.

To compare to shuffled data, we repeated the same fitting procedure described above 5 times, but randomly shuffled the evidence values across trials of the same sign-of-evidence for each position.

**Nonlinear evidence decoding in HPC.** In Supplementary Fig. 11, we use a nonlinear manifold method[103] to decode evidence from the neural population across timepoints, unlike the linear evidence decoding models described above. On each fold of a five-fold cross-validation, trials were assigned to training or test sets. To maximize the amount of data used to build the embedding, both correct and incorrect trials were used. We then fit a single CEBRA manifold[103] (https://cebra.ai/) embedding of dimension $d$ to the full trial trajectories in the training set for values of $d$ from 2 to 7, using the following parameter settings: model_architecture = "offset10-model", batch_size = 512, learning_rate = 3e-4, temperature_mode = "auto", output_dimension = d, max_iterations = 10,000, distance = "cosine", conditional = "time_delta", device = "cuda_if_available", time_offsets = 10. At each point, we provided the current evidence value as the label for the neural data. We trained a $k$ nearest neighbors decoder ($k = 3$) to predict the evidence level from the embedded training set. We then embedded the test set into the CEBRA manifold and used the decoder to predict evidence from the embedded test set. We then found the Pearson correlation between the predicted and actual evidence levels on the test set.

To compare performance to chance when controlling for choice, we repeated this process for 5 shuffles of the data in which the neural data was preserved, but the evidence labels were permuted across trials within trials of the same behavioral choice.

**Population-averaged tuning curves.** In Fig. 6L, M, we calculated the population-averaged response across all significantly evidence-selective neurons as follows. For each neuron, we calculated the average firing rate in each position-evidence bin. If a bin had fewer than 3 observations for a particular neuron, that neuron was not included in the population average for that bin. To combine data for neurons that preferred positive evidence with data from neurons that preferred negative evidence, for each neuron, we defined its preferred-evidence as the sign of the neuron's $\mu_e$ value and remapped the evidence axis to be in terms of preferred evidence rather than absolute evidence (by multiplying the evidence values by −1 if the preferred tuning was for leftward values of evidence instead of rightward). We then took the average across all neurons in each bin of position by preferred-evidence. Bins which were not sampled three or more times on at least 10% of the sessions for a particular brain region were not plotted.

## Reporting summary
Further information on research design is available in the Nature Portfolio Reporting Summary linked to this article.

## Data availability

The ACC, DMS, and HPC data have been uploaded to Figshare (https://doi.org/10.6084/m9.figshare.30921038)[151]. Previously published data from RSC is available at (https://doi.org/10.5061/dryad.cvdncjt53)[152]. Source data are provided with this paper.

## Code availability

Code for the simulation of all models and the data analysis is available at https://github.com/lindseysbrown/evidence_accumulation_through_sequences[153].

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

## Acknowledgements

We thank Sue Ann Koay for her useful discussions about the retrosplenial cortex dataset. We thank Shizhe Chen, Tim Hanks, Ben Lankow, and Avinash Baidya for useful discussions about this project. Funding was provided by NIH T32MH065214 (L.S.B.), F32MH132179 (L.S.B.), NIH-NINDS BRAIN Initiative 5U19NS104648 (L.S.B., D.W.T., C.B.D., I.B.W., and M.S.G.) and 1U19NS132720 (L.S.B., D.W.T., C.B.D., I.B.W., and M.S.G.), K99DA053388 (E.H.N.), Sloan Swartz Foundation (L.S.B.), C.V. Starr Fellowships (J.R.C., M.S.), Burroughs Wellcome Fund CASI awards (L.S.B., M.S.), and the Howard Hughes Medical Institute (I.B.W.). This manuscript is the result of funding in whole or in part by the National Institutes of Health (NIH). It is subject to the NIH Public Access Policy. Through acceptance of this federal funding, NIH has been given a right to make this manuscript publicly available in PubMed Central upon the Official Date of Publication, as defined by NIH.

## Author contributions

L.S.B., I.B.W., and M.S.G. conceptualized the model and designed the analyses. L.S.B. wrote the code and performed the data analysis. J.R.C., S.S.B., E.H.N., and M.S. collected data and curated neural recordings. D.W.T., C.D.B., and I.B.W. supervised experimental data collection. I.B.W., M.S.G., supervised the project and data analysis. L.S.B., I.B.W., and M.S.G. wrote the initial draft of the manuscript. All authors provided feedback on the manuscript.

## Competing interests

The authors declare no competing interests.

## Additional information

**Supplementary information** The online version contains Supplementary material available at https://doi.org/10.1038/s41467-026-70267-9.

