## [Transparent Peer Review File · Nature Communications]

Neural circuit models for evidence accumulation through choice-selective sequences

Corresponding Author: Dr Mark Goldman

Version 0:

Reviewer comments:

Reviewer #1

(Remarks to the Author)

In this manuscript, the authors developed two candidate circuit model classes to explain sequential (rather than persistent) firing patterns of neurons in an evidence-accumulation decision-making task. Such responses have been observed in many recent experiments. The models combined components from previously developed sequence models and evidence accumulation models to achieve sequential firing and evidence accumulation at the same time. The study highlights differences in neural tuning for evidence accumulation between the two models and describes how perturbation might additionally tell them apart. The study then compares tuning predictions to recorded calcium imaging data. Different forms of tuning are found in different areas, potentially suggesting that these areas might be using the different model classes.

Overall, the structure and logic of the manuscript were both clear and the supplementary material contains useful workups of key features of the models, such as the stability of differences in amplitudes of the two chains across positions in the absence of external cue inputs. That being said, the models are not particularly novel, in the sense that they are first in essence a composition of previously known models in the literature, and second more of a specific instantiation of two already existing conceptual ideas for how evidence might be accumulated in sequences, competing chains (directly inspired from the data) and a progression in a space of evidence (as suggested by the co-authors in previous work and referred to in the discussion). It is interesting that different brain regions appear to have different tuning, but the non-monotonic tuning is mostly in hippocampus where previous work by one of the authors has shown place-cell like encoding for progression in non-spatial spaces (auditory) and thus may be less surprising. Nonetheless, having these models as concrete anchor points for these ideas is a useful contribution to the field.

Questions:

1. The results from simulation and data analysis were always shown as an average across trials (and in places as averages across neurons as well). I understand that the task structure is more varied than some decision-making tasks in the way that visual stimuli are presented as Poisson sequences, but given that responses that are shifted in time average poorly I was surprised that there was almost no single-trial population level analysis. Both in general given previous literature, and since in classical attractor models, the nature of drift can be very informative of the underlying dynamics. Moreover, single trial analysis could perhaps show directly what are the changes in neural activity that happen when the same evidence is presented in two different positions, or the same position is traversed with or without evidence. In addition, it might give a better sense of how the models overall match the data.
2. The fitting of firing rate as a function of position and evidence and the fitting of evidence tuning curves around the most active position were both done by solving a least squares problem, i.e., by minimizing the sum of squares of residuals. However, the metric used by the authors to evaluate the fit of the models was the correlation coefficient between the predicted firing rates and true firing rates. The two metrics are clearly related and likely highly correlated, but was there a reason for this switch between the optimized quantity during model fitting (minimized residuals) and the actual evaluation of models (correlation coefficient)? How will the results change (e.g. Fig 5) if residual was used as the evaluation metric of models?
3. In the last results section "Progression in Graded Evidence Versus Binary Choice Tuning..." I may have misunderstood the analyses and my suggestion may have actually been performed, but I thought a helpful analysis would be comparing the classification (evidence-tuned vs choice-tuned) of a neuron with analysis in trials where the accumulated evidence is close to zero (i.e. the numbers of left and right stimuli are similar). It seems to me that neurons that are choice-tuned would still be able to predict the animal's choice reliably in these trials, but neurons whose choice tuning is inherited through the

association between evidence and choice would perform poorly. Was this the analysis described in the last paragraph? I think not but am unsure.

4. Many example evidence-tuned neurons visually seem to be very choice selective, especially when the neurons are selective at the delay regions (e.g. Extended Data Fig 6, examples 27, 28, 29, 30, 31, 33, 34, 36). The large difference between negative and positive evidence (i.e., choice) are clearly visible, but it is hard to see graded firing rates under different evidence levels. Is this just a visualization issue or are they only weakly selective to evidence?

Minor comments:

1. Are the evidence tuning curves from data (Fig. 5B) informative of the difference between the coupled version of the competing chains model (Fig. 3D) and the uncoupled version (Extended Data Fig. 2D)?
2. Would it be possible to directly fit the parameters of the competing chains model or the position-gated bump attractors using the experimental data?
3. Reference 119: I believe the authors accidentally referenced a change in a manuscript and not the actual manuscript.
4. In figure 1D, H, L I found the y-label "density" for what is described as a histogram in the legends a bit confusing, in particular as the scale changes five-fold from the left panel to the right panel.

(Remarks on code availability)

Reviewer #2

(Remarks to the Author)
Comments to Authors

The current study presents a modelling effort to describe evidence accumulation in neural networks where neurons are activated sequentially as a rodent traverses an environment. Two types of models are considered (monotonic and 'bump-shaped' tuning). The authors analyze data from new and previously published datasets collected from mice performing the 'towers task' from several brain areas — in which such sequential activity is present — and study which model type corresponds more closely to the neural activity in each brain area.

The data analysis is nice, although the idea of joint evidence and spatial encoding is not novel. The modelling effort feels a bit raw.

Major Comments

1. Weak link between model and neural activity. The authors construct models satisfying a series of desiderata for linear evidence accumulation in sequential models. However, the evidence that this phenomenon is taking place in the recordings is fairly limited.

In the areas showing 'monotonic tuning' (RSC and ACC) neurons seem to largely represent the best current categorical guess (Figs. 5 and 6). Although the authors argue that linear evidence accumulation transitions to binary choice encoding (lines 532-533) their Fig. 6D,F shows that, at best, the small fraction of neurons that have non-zero regression coefficients for either variable is similar. Then only binary choice becomes significantly represented later in the corridor. Furthermore, even at these early positions we are talking about ~ 2% of neurons, and also, at these early positions distinguishing between binary choice and graded evidence becomes more difficult (in the limiting case there only one tower has been seen they cannot be distinguished).

In HPC, convincing evidence is shown that a significant proportion of neurons have 'bump shaped' tuning curves for evidence, but this is also fairly indirect evidence that the population as a whole is tracking the running accumulated evidence in the task. There are certainly more direct ways to probe if the HPC is representing instantaneous accumulated evidence or not.

Thus, the bulk of the constraints that are used to develop the models derive from functional considerations that are not experimentally demonstrated. If, for instance, the data in Figs 1,5-6 was presented first, it would be more salient that the modelling in Figs 3,4 only loosely maps to the data.

What the data convincingly demonstrates is a dichotomy in terms of the shape of tuning curves to evidence, but this fact, without the requirements for graded evidence accumulation, provides a much weaker constraint on the mathematical models. If the authors wish to make a tight link between data and modelling, less constrained models (presumably a larger model class) should be considered. If the authors wish to explore possibilities for modelling sequential graded accumulation of evidence with a mostly theoretical motivation, then the theoretical and experimental aspects of the manuscript become less tightly associated. It would need to be made explicit that the data relates to these models fairly indirectly, and instead bears on a much larger model class, which seems to partly defeat the purpose of having the data and models together as it is done in the present version.

2. Schematic and somewhat unrealistic modelling approach. While I appreciate the effort of translating the desiderata into specific mathematical implementations, the models presented seem somewhat artificial. A large number of tuning conditions need to be satisfied. The treatment of the all-or-none position signal is also quite artificial. The data in Fig1 shows graded and heterogeneous spatial 'tuning curves' and connected neurons will interact throughout the corridor, not just through the

'position gate'.

Some specific issues I had with the modelling:

2.1 I might be mistaken, but it seems to me like the formula between lines 113 and 114 in the Supp. Text is incorrect? The upper limit of the integral is P_0 , but it should be quantity with dimensions of time. Presumably P_0/speed , i.e., the time during which the position gate is open? Given the values of these parameters, $a \cdot P_0/\text{speed} = 1 \text{ Hz} \cdot 20 \text{ cm} / 50 \text{ cm/s} = 0.4$, so that each position is actually integrated quite partially. As the authors point out, full integration would require this quantity to be $\gg 1$.

2.2 Why does the neural activity profiles in figs 3C, 4C look so discontinuous along space? If the decay rate of activity is 1Hz and the mice runs at 50cm/sec (see comment below on speed), then graded decay should be evident on a 300 cm region.

2.3 What is the role of speed on the activity of the network? If the position gate is fixed integration of evidence will proceed more slowly for faster mice? Is this reasonable?

2.4 What is the shape of the dynamics of the sum mode in the competing chains model? Are there any qualitative features that could be compared with the data? The global activity of the network (and/or evidence-tuned neurons is readily accessible).

Overall, readers will benefit from a more thorough characterisation of the models. More generally, while the models seem like a good first step, they seem too schematic and unrealistic for a finished product, specially given the fact that they are structurally unstable and the adding both functional heterogeneity (present in the data) and parameter heterogeneity (present in the brain) is not simply a matter of cosmetics, but rather represents an important challenge to the desired model function.

Other comments:

1. Fig 2 seems expected. Maybe it could go to the supplementary info.

2. The optogenetic perturbation simulations in Figs 3 and 4, while interesting, take a lot of space and are not pursued experimentally. The authors might consider again moving them to the supplementary information and using that space to describe more precisely how the models work (in a sense deconstructing panel C by showing temporal profiles of synaptic currents from different sources etc), effect of different parameters, etc

3. In lines 571-2 the authors speculate that the position signal (which in the model is totally external), might come place cells in the hippocampus, but the authors are actually modelling the hippocampus. This seems strange.

(Remarks on code availability)

Version 1:

Reviewer comments:

Reviewer #1

(Remarks to the Author)

The reviewers responded in detail to my concerns and have performed a substantial amount of work in the revision. In my opinion, some of the responses, such as the addition of a comparison of correct and incorrect trials were useful and added to the paper. Others, like the clarification of regarding the use of "evidence-tuned" were important clarifications. However, my main concern, adding more single trial analyses, was less well answered. My comment was that there should be aspects of single-trial dynamics that are quite different between the two types of models. Just as the authors describe perturbation experiments that would better differentiate between the models, there should be single trial analyses that do the same. For instance, the authors write that (starting line 384): "This is because the random walk of the bump location as evidence is accumulated leads to a greater range of evidence levels being observed at later positions in the trial; thus, on average across trials, there are more different neurons active at later positions, with each neuron corresponding to a different possible evidence level". This an example property that could have been better probed with a single trial analysis. I don't believe this is the best example, I just mention it as an example of some single-trial analysis that makes expected model properties (sharper bumps) more apparent. My suggestion was for one or more analyses along such lines that clarify and then contrast properties of the two suggested models. The population analyses performed instead were more focused on population-level versions of overall properties. Analysis (i) shows that evidence can be decoded (which is not surprising given the single neuron results) analysis (ii) looks for sustained responses, which unless I am mistaken have to exist in both models and analysis (iii) is more of a visualization that better reveals graded encoding. These are all interesting, but do not address my specific comment which is on single trial dynamics that differentiate the two models. I appreciate the authors' response that drift is tricky and I fully appreciate that analysis and some others can't be done given the recordings available. My comment referenced that as one specific example of a broader set of analyses which I believe would have strengthened the paper.

That being said, I believe the paper is suitable for publication.

Two more minor comments: 1. I agree with reviewer 2 that Figure 2 is not particularly useful and can be sent to supplementary. 2. I personally find the model schematics very difficult to parse. To me the equations and verbal descriptions were more useful. While I imagine the authors have already dedicated some thought to this, it would be nice if they could find some better schematics (which can be for instance tested by asking people not directly involved with the study what they infer from the schematics).

(Remarks on code availability)

Reviewer #2

(Remarks to the Author)

The authors have largely satisfactorily addressed my concerns.

(Remarks on code availability)

Reviewer #3

(Remarks to the Author)

I have been brought in as an additional reviewer following missing responses from R2 in this review process. After reading the responses provided to R2's concerns, the authors have thoroughly and carefully answered all outstanding questions. R2's general critique was slightly vague at times, yet the authors have gone above and beyond, congratulations.

(Remarks on code availability)

We thank the reviewers for their feedback. Reviewer 1 praised the modeling as “a useful contribution to the field” and found that “the structure and logic of the manuscript were both clear,” the data analysis results were “interesting,” and the supplementary material was “useful.” Reviewer 2 also found that the “data analysis is nice” and “convincingly demonstrates ... a dichotomy in terms of the shape of tuning curves to evidence.”

In addition, both reviewers provided important suggestions to improve both the modeling and analysis aspects of the paper. The two main criticisms of the work were, first, that our data analysis did not clearly show a representation of graded evidence, as assumed in the models and, second, that the models did not recapitulate a key feature of the data, in particular the presence of a binary choice population that increased in size as the animal traversed the maze. We have addressed this constructive feedback by:

1. Providing a more complete analysis of evidence representations in the brain regions that we analyze.

In the broadest sense, we consider evidence tuning to be any predictable response in firing as a function of evidence, which can incorporate the (i) graded, monotonic evidence tuning of the chains-based models, (ii) the non-monotonic tuning of the position-gated bump attractor, and (iii) choice tuning, since choice will be correlated with evidence on correct trials. However, for a brain region to be a candidate integrator, it must have decodable representations of *graded* evidence. In the original version of the manuscript, we characterized the evidence tuning of individual neurons in each region and presented correlational analyses (single-neuron encoding models) that demonstrated that some neurons in ACC and RSC had graded encoding of evidence. We now both improve upon and add to these analyses in the following ways:

- **Revising our individual neuron evidence vs. choice encoding analysis models to explicitly separate behavioral choice from evidence.** We have now updated our single-neuron evidence vs. choice encoding analysis to more directly account for behavioral choice by fitting our model to evidence and choice (rather than sign-of-evidence) on both correct and incorrect trials (rather than just correct trials). Incorporating incorrect trials allows us to better distinguish cells that respond to behavioral choice from those that respond to evidence, since the two are highly correlated on correct trials. This modification revealed a much larger portion of cells that responded to linear evidence beyond behavioral choice.
- **Building evidence decoders to show that the regions we analyze carry graded evidence information.** We now demonstrate that evidence can be linearly decoded, even after controlling for the current best choice, from the populations in ACC and RSC, lending further support that these regions have evidence representations. In addition, our decoding analysis shows that there is a transition from evidence coding to an increasingly choice-like representation later in the maze, where decoding of evidence falls to chance levels in the delay. This observed transition is consistent with recent work suggesting that neural dynamics change at the time of decision commitment (Luo et al. 2023).

Separately, we have added an analysis showing that evidence could be decoded from the HPC by utilizing a nonlinear manifold.

- **Using a cue encoding model to uncover signatures of evidence integration.** We fit a cue encoding model to the population activity that models neural activity as a linear sum of a temporal kernel associated with each cue (“cue kernel”), and found that cue response kernels in both ACC and RSC had the causal, sustained, approximately step-like responses to the appearance of cues. This signature is a necessary feature of an integrator.
 - **Incorporating single-trial analyses to further probe evidence responses in these regions.** In addition to single-trial decoding analyses and population cue encoding model fits, we performed additional analyses of activity on individual trials. We showed that average trial trajectories showed graded patterns based on the final evidence during the cue region, which then collapsed to choice-like representations in the delay region, again consistent with a transition from evidence coding to an increasingly choice-like representation later in the maze.
- 2. Making the connection between the data and the models more explicit.** Our original models were built to integrate evidence, but our data analysis revealed that in addition to cells tuned to evidence, there is a large population of cells that respond to behavioral choice, especially toward the end of the maze. We now include two expanded circuit models that instantiate hypotheses for how such choice-tuning emerges as the animal traverses the maze, as well as a model for how position encoding may emerge from an integration of velocity.
- **Explicitly modeling a population of choice cells that can read out the representations of both models.** One of the major differences between the predictions of the chains-based models and the patterns observed in the neural data in regions with monotonic evidence tuning was the transition to a more choice-like representation later in the maze in the neural data. We now explicitly model two ways this could occur (Fig. 7): first, we model a population of choice-cells that respond to the difference in the two chains at different positions and show that, when there are increasingly more of these cells at later positions, we can reproduce the experimental, population-data results of Figure 6L,M. Alternatively, we show that the shift towards a more binary choice tuning can occur within the integrator if it is slightly unstable, as suggested for other accumulation of evidence tasks (e.g., Wong & Wang, 2006).
 - **Incorporating a mechanistic model of velocity integration into our bump-based model to allow position and evidence to be jointly integrated.** In our previous version of the manuscript, we assumed that the position gating signal that controls which neurons are active derives from the observed spatially selective cells that do not encode evidence (these cells are now shown in Extended Data Figure 1). However, it is also possible that position and evidence are integrated within the same population. We now include in the manuscript an alternative, planar bump attractor model that integrates position along one axis and evidence along the other (Extended Data Fig. 3). This velocity-to-position

integration mechanism could also be used to derive a position gating signal, separate from the evidence accumulation mechanism.

In addition to the major improvements outlined above, we have also made smaller updates based on reviewer feedback, including extending our discussion of the robustness of our models and demonstrating robustness to smoothness and heterogeneity in the position gate. The reviewers' comments (black italics) and our individual responses (blue) to each point are included below.

Overall, our work contributes two models that provide potential explanations for the choice-selective sequences that are present across regions during evidence accumulation and the different shapes of tuning curves in the hippocampus and in cortical regions. The additional analyses suggested by the reviewers helped to further characterize the differences across regions. We now show how these models can be modified to account for cells that respond to binary choice in the data. That being said, we note that one of the biggest experimental questions in the field is where the integration in evidence accumulation based decision making tasks is occurring, as it is difficult to distinguish the locus of integration from a readout of an upstream integrator – we hope that the predictions of our mechanistic models lay the groundwork for perturbation experiments that can further probe the coordination of evidence accumulation across the brain in cases where sequential, rather than persistent, neural activity is observed.

REVIEWER COMMENTS

Reviewer #1 (Remarks to the Author):

In this manuscript, the authors developed two candidate circuit model classes to explain sequential (rather than persistent) firing patterns of neurons in an evidence-accumulation decision-making task. Such responses have been observed in many recent experiments. The models combined components from previously developed sequence models and evidence accumulation models to achieve sequential firing and evidence accumulation at the same time. The study highlights differences in neural tuning for evidence accumulation between the two models and describes how perturbation might additionally tell them apart. The study then compares tuning predictions to recorded calcium imaging data. Different forms of tuning are found in different areas, potentially suggesting that these areas might be using the different model classes.

Overall, the structure and logic of the manuscript were both clear and the supplementary material contains useful workups of key features of the models, such as the stability of differences in amplitudes of the two chains across positions in the absence of external cue inputs. That being said, the models are not particularly novel, in the sense that they are first in essence a composition of previously known models in the literature, and second more of a specific instantiation of two already existing conceptual ideas for how evidence might be accumulated in sequences, competing chains (directly inspired from the data) and a progression

in a space of evidence (as suggested by the co-authors in previous work and referred to in the discussion). It is interesting that different brain regions appear to have different tuning, but the non-monotonic tuning is mostly in hippocampus where previous work by one of the authors has shown place-cell like encoding for progression in non-spatial spaces (auditory) and thus may be less surprising. Nonetheless, having these models as concrete anchor points for these ideas is a useful contribution to the field.

We thank the reviewer for their careful reading of this manuscript, and for pointing out that the work provides a useful contribution. We also appreciate the constructive comments. The reviewer correctly notes that, within a position, the models reduce to well-established integrator models. Thus, the novelty of the modeling was extending these traditional models to transfer graded evidence across positions. In addition, we agree with the comment from the reviewer that these models serve as useful anchor points. Indeed, as such, they suggested key analyses to identify differences between brain regions, which in turn revealed that what were thought to be similar choice-selective sequences across regions had different evidence tuning schemes (and thus signatures of the different models). Beyond this insight into our data, we believe that our models also provide a more general conceptual insight beyond previous models by introducing a gating system that allows for the transfer of graded information between positions, which we explicitly formulate here. In ongoing work, we are demonstrating how a very similar formulation can explain recent primate data from the Shadlen lab during an evidence accumulation task in the presence of eye movements (So & Shadlen, 2022), where in that case neural signals related to the eye position serve as the gate.

Questions:

1. The results from simulation and data analysis were always shown as an average across trials (and in places as averages across neurons as well). I understand that the task structure is more varied than some decision-making tasks in the way that visual stimuli are presented as Poisson sequences, but given that responses that are shifted in time average poorly I was surprised that there was almost no single-trial population level analysis. Both in general given previous literature, and since in classical attractor models, the nature of drift can be very informative of the underlying dynamics. Moreover, single trial analysis could perhaps show directly what are the changes in neural activity that happen when the same evidence is presented in two different positions, or the same position is traversed with or without evidence. In addition, it might give a better sense of how the models overall match the data.

The reviewer correctly points out the original version of this manuscript was very limited in single-trial population analysis, largely for the reason described that the timing of the cues differs from trial to trial. Nevertheless, prompted by the reviewer's comment, in addition to single-neuron analyses of trial-averaged data (see response to comment 3), we have now performed a number of single-trial analyses to evaluate evidence responses in the population. The analyses performed were: (i) single-trial decoding analyses, (ii) cue encoding models fit to the set of single-trial time series data (rather than to trial-averaged data), and (iii) single-trial trajectories, shown as a set of separate averages for different levels of evidence. In our expanded figure, we show some example trials, but many single trials were quite noisy and thus

not ideal for displaying trends; thus, as shown below, we focused most of our analysis plots on the averages of the single-trial results.

Before presenting these new analyses, regarding “what are the changes in neural activity that happen when the same evidence is presented in two different positions, or the same position is traversed with or without evidence,” we want to clarify that our original joint position by evidence tuning curve analysis also provided some insight into this question. We showed that single neurons are tuned to position (Figs. 1 B-E, 5A,E,I), such that individual neurons only respond at specific positions, including only to evidence at those specific positions. Similarly, these tuning curves show that for many neurons, firing rate is strongly evidence-dependent at the best position (Fig. 5B,F,J). However, the population analyses of evidence representations detailed below provide further insight into these questions.

(i) Single-Trial Evidence Decoding Analyses

We analyzed whether evidence could be decoded on single trials from the population. We found that the population could decode evidence above chance (as defined by a current-best-choice matched null distribution). Decoding accuracy fell to chance levels during the delay, supporting the observed transition to a more choice-like representation later in the maze (Fig. 6 I-K, Extended Data Fig. 13 G,H). We note that this transition from evidence-encoding to choice-encoding is consistent with both our other analyses in the manuscript and a similar, recent finding that neural dynamics shift from an evidence accumulation to a decision commitment regime (Luo et al. 2023), so our neocortical recordings may be suggestive of such a commitment process (Johnson et al. 2017) occurring in the latter portions of the maze.

Fig. 6 I, J, K (I) Top: Linear evidence decoding model at each position. Bottom: Example trials with actual (solid) and decoded (dashed) evidence in ACC. (J) At each position, the cross-validated correlation between actual evidence and predicted evidence from a linear population decoder in ACC on correct right evidence ($e \geq 0$) trials (red) and left evidence ($e \leq 0$) trials (blue), compared to shuffle (grey). Error bars indicate s.e.m. across sessions. (K) Same as (J) but for RSC.

(ii) Population-based Cue Encoding Models

We also asked whether the population responses showed sustained cue responses that are necessary for integration. We did this by fitting a cue encoding model to the time-series data. In this model, cues are convolved with a response kernel to predict the difference of right-preferring and left-preferring population activity. Our model was given by

$$f_{\text{pop}}(p) = \beta_0 + \sum_{l \in L} (K_L * l)(p) + \sum_{r \in R} (K_R * r)(p)$$

where f_{pop} is the difference in the average activity of the right evidence-preferring cells and the average activity of the left evidence-preferring cells at position p , L is the set of left cues, R is the set of right cues, and K_L and K_R are the cue response kernels to left and right cues respectively.

We found that in both ACC and RSC, the population showed sustained responses of opposite sign to left and right cues that are consistent with and necessary for integration (Fig. 6 A-E).

Fig. 6 A-E (A) Schematic of the cue encoding model, where each left and right cue is convolved with a left or right cue response kernel, K_L or K_R respectively, to predict the population activity. (B) Example single-trial population activity (black) compared to the model predicted activity (dashed gray) in ACC. (C) Same as B but for RSC. (D) Cue kernels fit from the cue encoding model to right cues (red) and left cues (blue). (Average $r^2 = 0.26$ across sessions.) (E) Same as D but for RSC. (Average $r^2 = 0.20$ across sessions.)

The reviewer also raises an interesting question about population responses to evidence at different positions across the trial. Thus, we additionally fit cue encoding models in which there were different kernels for cues presented at different evidence levels or positions (Extended Data Figure 13A-D). The cue kernels were smaller for larger absolute evidence, consistent with the transition from evidence to choice representations as the maze progresses (Extended Data Figure 13A,B). Interestingly, the cue response kernels decreased less as a function of position in the maze (Extended Data Figure 13C,D; note nonzero kernels even for later positions), suggesting that the transition from evidence-encoding to choice-encoding activity may be due to animals committing to decision with larger levels of evidence, rather than being due solely to just being at a later position in the maze.

Extended Data Figure 13A-D. (A) Change in the difference in activity of the right and left population in ACC following a left cue (blue) or right cue (red) when the current absolute value of evidence is low (light colors) to high (dark colors). (Average $r^2 = 0.29$ across sessions.) (B) Same as A but for RSC. (Average $r^2 = 0.21$ across sessions.) (C) Change in the difference in activity of the right and left population in ACC following a left cue (blue) or right cue (red) when the cue appears in the early (light colors) to late (dark colors) cue region. (Average $r^2 = 0.27$ across sessions.) (D) Same as C but for RSC. (Average $r^2 = 0.20$ across sessions.)

(iii) Population Trajectories for Trials of Different Evidence Levels

In Extended Data Figure 13D,F, we have added an analysis to show the evolution of the neural response over positions for different final evidence levels (for correct choice trials). Unlike the tuning curve analysis that was already included in the manuscript (Fig. 5B,F,J, Extended Data Fig. 6), these averages are performed by trial (rather than by evidence level at a fixed position, as in Fig. 5), such that entire trial trajectories are preserved. This analysis uncovers a similar pattern to our average population tuning curve analysis (Fig. 6L,M), where we observe graded responses, especially early in the maze.

Extended Data Figure 13E,F. (E) Trial-averages of the difference of mean activity in the left and right populations of active cells with significant evidence coefficient in ACC for trials with different final evidence levels (indicated by the color bar). (F) Same as E but for RSC.

Unlike what we presented in our initial submission, these analyses preserve single trials prior to averaging. However, we acknowledge that this analysis does involve trial averaging that can obscure individual trial dynamics. As the reviewer notes, one measure that has been shown to distinguish dynamics in single-cell data where cells are persistently active over the trial is to test “the nature of the drift” (Latimer et al. 2015, Zoltowski et al. 2019). A ramping model would predict that, in the presence of noise, the variance around the mean trajectory for a given set of cues grows linearly over the course of the trial for all evidence levels. This style of analysis is effective in data where cells show persistent firing across the trial but is not easily applied here due to the discrete nature of the evidence stream and the sequential structure of cellular activity. Specifically, for single cells, on each trial, the neuron’s tuning is modulated by position, so that variance across positions is not a meaningful measure. Similarly, at the population level, different numbers of cells are tuned to different positions, obscuring variance that results from accumulation processes and variance resulting from changing cell counts. In summary, tests for this type of drift are not readily applied to our data due to the high position selectivity observed in our data. Despite this limitation, we hope that the new set of analyses described above provide convergent support for the presence of a graded evidence encoding component in the population during the cue period.

2. The fitting of firing rate as a function of position and evidence and the fitting of evidence tuning curves around the most active position were both done by solving a least squares problem, i.e., by minimizing the sum of squares of residuals. However, the metric used by the authors to evaluate the fit of the models was the correlation coefficient between the predicted firing rates and true firing rates. The two metrics are clearly related and likely highly correlated, but was there a reason for this switch between the optimized quantity during model fitting

(minimized residuals) and the actual evaluation of models (correlation coefficient)? How will the results change (e.g. Fig 5) if residual was used as the evaluation metric of models?

Thank you for this comment. In the initial manuscript, we used the correlation between the model predicted firing rates and the true firing rates in comparison to correlations in pseudosessions to define whether a neuron had significant evidence tuning, but the reviewer correctly points out that it is more consistent with our fitting method to select neurons as evidence selective based on MSE in comparison to MSEs in the pseudosessions. We agree and have now updated the figures, text, and methods to use MSE.

We note that this change in goodness of fit metric led to minimal change in our results, as we found almost perfect overlap between the cells identified as significantly evidence-tuned based on residuals and those identified as significantly evidence-tuned based on correlation, as seen in the almost complete overlap in the Venn diagrams below (Fig. R1.1). However, the small fraction of non-overlapping neurons as well as the imperfect mapping between the two measures does create minor changes in the data used for our figures and the percentages of cells reported as having significant evidence tuning in each region.

Fig. R1.1 Overlap of cells showing significant evidence tuning. Venn diagrams showing the number of neurons identified as having significant evidence tuning defined based on (i) the correlation (pink) or (ii) the mean squared error (MSE; green) between the model predictions and actual firing when compared to pseudosessions. Overlap is seen in brown. Numbers on the plots indicate the number of neurons in each category.

3. In the last results section “Progression in Graded Evidence Versus Binary Choice Tuning...” I may have misunderstood the analyses and my suggestion may have actually been performed,

but I thought a helpful analysis would be comparing the classification (evidence-tuned vs choice-tuned) of a neuron with analysis in trials where the accumulated evidence is close to zero (i.e. the numbers of left and right stimuli are similar). It seems to me that neurons that are choice-tuned would still be able to predict the animal's choice reliably in these trials, but neurons whose choice tuning is inherited through the association between evidence and choice would perform poorly. Was this the analysis described in the last paragraph? I think not but am unsure.

We thank the reviewer for this question about the differences between evidence and choice cells. This prompted us to take a more straightforward approach to comparing evidence and choice by revising our original single-neuron encoding analysis to explore evidence vs. behavioral choice tuning of the neurons instead of responses to linear evidence vs. sign-of-evidence as in the original manuscript. Most fundamentally, we realized that we could more directly test whether neurons are tuned to behavioral choice by using incorrect trials to see whether a neuron is following the choice. Specifically, we now use both correct and incorrect trials in our evidence vs. choice single-neuron analyses to explicitly test for the dependence of individual cell firing at each position on evidence and behavioral choice. This analysis revealed larger proportions of cells that encoded evidence beyond choice compared to the more minor distinction in the shape of evidence tuning tested for in the previous single-neuron encoding model.

Fig. 6 F, G, H (F) Evidence vs. choice single-neuron encoding analysis. Top: regression equation. Bottom: schematics of the firing across positions and evidence levels for a purely evidence encoding cell and a purely choice encoding cell on correct trials. **(G)** For each position, the average fraction of active neurons in each session in ACC (defined as neurons with fit position mean within one position standard deviation of the animal's location) with significant coefficients for evidence (crimson) or choice (lime) in the choice vs. evidence encoding model, based on the F-statistic. Error bars indicate s.e.m. across sessions. The horizontal dashed gray line indicates chance level (1%). **(H)** Same as (C) but for RSC.

This more straightforward approach to differentiate evidence and choice cells relies on incorrect trials, rather than needing to consider trials near zero evidence which are rare in our dataset. For the reviewer's reference, we did perform the suggested analysis, and as the reviewer expected, we found that in the original single-neuron encoding model, the accuracy of individual cells in decoding sign-of-evidence depended significantly on the magnitude of evidence for cells with significant evidence coefficients (ACC: $p < 0.01$, RSC: $p < 0.01$; Wald Test with t-distribution of test statistic for slope of a linear regression), but not for those cells with

only significant sign-of-evidence coefficients (ACC: $p = 0.09$, RSC: $p = 0.11$; Wald Test with t-distribution of test statistic for slope of a linear regression).

4. Many example evidence-tuned neurons visually seem to be very choice selective, especially when the neurons are selective at the delay regions (e.g. Extended Data Fig 6, examples 27, 28, 29, 30, 31, 33, 34, 36). The large difference between negative and positive evidence (i.e., choice) are clearly visible, but it is hard to see graded firing rates under different evidence levels. Is this just a visualization issue or are they only weakly selective to evidence?

We thank the reviewer for this question, which points to an overloading of terminology that was not well explained in our original manuscript. When we describe a cell as “evidence-tuned”, this refers to cells that show any significant modulation by evidence when compared to the pseudosession. Thus, evidence-tuned cells include three classes of interest to our study: (i) non-monotonically evidence-tuned cells, (ii) graded evidence-tuned cells, and (iii) choice-tuned cells. The choice-tuned cells respond to the sign-of-evidence on correct trials, but in a binary manner, so are still correlated with evidence on these trials. The example cells that the reviewer points to are best classified visually as cells of type iii, which are tuned to choice with limited (if any) graded information. The qualitative pattern observed by the reviewer (more cells that appear choice-tuned in the delay region) is observed in the population tuning curves (Fig. 6L,M) and is precisely what motivated the analysis of evidence beyond choice vs. choice beyond evidence tuning that we quantified in our single-neuron models (Fig. 6F-H).

To make this clearer in the updated manuscript, we have edited our discussion at the beginning of the section “*Change in Balance of Graded Evidence Versus Binary Choice Tuning Across Positions in ACC and RSC*”:

“While the monotonicity of the evidence tuning of individual cells observed in ACC and RSC is consistent with the competing chains models, a candidate integrator must represent graded evidence beyond monotonic, step-like evidence tuning driven by choice. Since the units of our models should not be thought of as individual cells, but rather populations of neurons with similar position tuning, the graded representation predicted within our model units could emerge from either individual cells having graded representations or individual cells stepping on at different evidence thresholds such that population averages are graded¹⁰⁴. Therefore, we examined whether the regions dominated by monotonic tuning for evidence contained representations of evidence in the population beyond binary choice tuning, using both single-cell and population-level analyses.”

Minor comments:

1. Are the evidence tuning curves from data (Fig. 5B) informative of the difference between the coupled version of the competing chains model (Fig. 3D) and the uncoupled version (Extended Data Fig. 2D)?

We thank the reviewer for this question about the differences between the two versions of the chains based models. The differences in the shapes of the tuning curves in Fig. 3E (previously panel D in the original submission) and Extended Data Fig. 2D should not be interpreted too strongly because both models can be modified to have either linear or saturated tuning curves.

The flattening (saturation) of the evidence tuning curves that is seen in Fig. 3E but not in Extended Data Fig. 2D is a result of the activity of one chain being brought below threshold by the dominant chain (lines 205-209). Because the mutual inhibition between the two chains is a (disinhibitory) positive feedback loop, and such positive feedback is what mediates the ability to accumulate evidence over time, the silencing of one chain when it goes below threshold leads to loss of positive feedback and saturation of the accumulating process. By increasing the amount of external input, $X(t)$, to each chain at the active position, we could remove the saturation in the mutually inhibiting chains model, resulting in linear tuning curves like those seen in the uncoupled chains model. However, we decided to use a parameterization of the model that showed the saturation since that appears more similar to tuning curves seen in the data.

Similarly, thresholding of low firing rates in the uncoupled model can occur if the initial firing rates are decreased. Saturation of higher firing rates does not occur in the current instantiation of the uncoupled chains model, but could easily result from biophysical, or more complex network-mediated, saturation effects that constrain maximal firing rates.

For these reasons, we do not interpret our tuning curve data as being informative of the differences between these variants of the chains-based class of models. Instead, we proposed optogenetic perturbations that would clearly distinguish between the two versions of the chains-based models as a much more direct approach. We do hypothesize that the mutually inhibiting chains may be more similar to the true neural circuitry due to the high levels of lateral inhibition within the cortex (Del Rosario 2025, Fan 2020) and for this reason presented it in the main text while leaving the other to the supplement.

We have expanded Extended Data Fig. 2 to explicitly show these possibilities, and we have added text to clarify that both the competing and uncoupled chains models can have either linear or saturated tuning curves:

- “This saturation is not a core feature of the model, however, and the non-saturating range of evidence levels can be modified by the amount of external input to each chain or the synaptic strength of the visual input connections (Extended Data Fig. 2J).” (lines 207-209)
- “As in the mutually inhibitory chains model, this model can either saturate or not depending on the strength of the synaptic input connections and assumptions about the dynamic range of the neurons (Extended Data Fig. 2K).” (lines 245-247)
- “Although these variant competing chains models cannot be reliably distinguished by their tuning since both models can be modified to have either linear or saturated tuning curves, ...” (lines 249-252)

Extended Data Figure 2J, K. (J) Tuning curves of individual neurons to evidence by finding the average activity for different evidence levels at the neuron's peak position, for left-preferring (blue) and right-preferring (red) neurons for a parameterization of the mutually inhibiting chains model that does not show saturation over the observed evidence range. (K) Same as (J) but for a variant of the uncoupled chains model with an upper bound imposed on firing.

2. *Would it be possible to directly fit the parameters of the competing chains model or the position-gated bump attractors using the experimental data?*

We agree that fitting model parameters to actual neural data is an exciting area, but outside the scope of this paper since the units of our model represent populations of neurons rather than individual cells. In our simulations, we chose parameters for our models that recapitulate typical firing rates of the data and that gave similar position widths to those in the data (Fig. 1B-E), as we have now made explicit in the Methods. This approach thus captures the average behavior of the population, for which our models were designed. For this reason, we believe that the additional population level analyses we have introduced are more informative for testing the predictions of our models than directly fitting model parameters to individual cells.

Nevertheless, our existing analyses do provide insight into some of the parameters of the models. In particular, our existing tuning curve analysis and fitting (Fig. 5) gives the magnitude of individual cell responses to cues in the chains based models (the integral of f_{left} (Eq. 1)) and the width of the tuning curves in the non-monotonic regions gives us a measure of the expected width of the excitatory connections in the position-gated bump attractor. Fitting parameters more precisely beyond the insights provided by these tuning curves would require additional perturbation experiments. Previous work in a well-studied neural integrator circuit has been able to successfully fit models of a similar nature by using targeted perturbation experiments and fitting the parameters to capture both the unperturbed and perturbed responses (Fisher et al. 2013). For each of our models, we outlined perturbation experiments that could probe the existence of the hypothesized circuitry (Fig. 3F,G, Fig. 4F-H, and Extended Data Fig. 2E,F) and that could similarly be used to fit parameters of our model. We have added this point to the discussion of the value of such perturbation experiments:

“We were able to compare evidence tuning in new and existing datasets to those predicted by the models. However, unlike targeted perturbations, correlational analyses do not provide causal evidence for the underlying computations in a region. Perturbations in the oculomotor neural integrator, where functionally distinct populations are conveniently anatomically separated, have successfully... allowed the parameters of a model of this system to be fit directly from neural responses in perturbed and unperturbed trials (Fisher et al. 2013).”

Critically, we do not expect all the cells in our data to be well fit by our models because of the large proportion of choice-tuned cells in our data and because our models were designed to accumulate evidence rather than readout choice. Therefore, we have added to the manuscript two new model variants that qualitatively reproduce the transition from evidence to choice tuning, through either the addition of choice readout cells or changing the parameters to make the integrator slightly unstable (Fig. 7). These expanded models better capture the transition observed in the data from graded evidence early in the maze to a choice-like representation later in the maze.

Figure 7. Simulated shifts towards choice tuning in the population. (A) Model schematic of a population of choice cells that readout from the accumulator and indicate if there is more left evidence (navy blue cells) or right evidence (dark magenta cells). The number of neurons in the choice readout population grows along the chain. (B) Tuning curves of the choice cells at their preferred position. (C) As in Figure 6, population average tuning curves of the entire population of evidence-tuned cells including both cells in the accumulator and the sign-of-evidence cells. Cross sections show a shift towards choice tuning with increasing position in the maze. (D) Schematic of the mutually inhibiting chains model with

unstable tuning, in which the strength of self-excitation increases with position along the chain. (E-F)
Same as B-C but for cells in the unstable mutually inhibiting chains model.

3. Reference 119: I believe the authors accidentally referenced a change in a manuscript and not the actual manuscript.

We thank the reviewer for their careful read of the bibliography and have appropriately updated this reference (Orhan & Ma, 2019).

4. In figure 1D, H, L I found the y-label “density” for what is described as a histogram in the legends a bit confusing, in particular as the scale changes five-fold from the left panel to the right panel.

We thank the reviewer for this question regarding the panels in Figure 5 D, H, and L. What we have plotted is the distribution of neurons across the normalized μ_e (left) and σ_e (right). We chose to plot this distribution as a density, rather than as absolute cell counts, to facilitate making comparisons across regions, since the number of recorded cells varies widely between regions. We have updated the figure legends to more accurately call these “density plots” rather than “histograms”. Note that the change in scale across the left and right panels is expected, given that the distributions of μ_e (left) and σ_e (right) are different for these different fit parameters. If preferred by the reviewer or editor, we could change to plotting this distribution in terms of absolute cell count with an axis that would vary between regions.

Reviewer #2 (Remarks to the Author):

Comments to Authors

The current study presents a modelling effort to describe evidence accumulation in neural networks where neurons are activated sequentially as a rodent traverses an environment. Two types of models are considered (monotonic and ‘bump-shaped’ tuning). The authors analyze data from new and previously published datasets collected from mice performing the ‘towers task’ from several brain areas — in which such sequential activity is present — and study which model type corresponds more closely to the neural activity in each brain area.

The data analysis is nice, although the idea of joint evidence and spatial encoding is not novel. The modelling effort feels a bit raw.

We thank the reviewer for their careful read of the manuscript and helpful suggestions to improve the connections between the modeling and the data analysis.

Major Comments

1. Weak link between model and neural activity. The authors construct models satisfying a series of desiderata for linear evidence accumulation in sequential models. However, the evidence that this phenomenon is taking place in the recordings is fairly limited.

Thanks for the question and we appreciate the concern (and hope, as described below, that we have mitigated the concern). We do want to step back, however, to highlight the main experimental finding that motivated the models and which we, first and foremost, wished to capture. This finding was the observation of sequential neural activity across the many different brain regions recorded during an accumulation-of-evidence based decision-making task (Fig. 1B-E, Extended Data Fig. 1). These data had previously been considered to be similar because of the similar choice-dependent, sequential structure shared by these regions (Koay et al. 2022, Nieh et al. 2021). Our models capture this structure while also performing the accumulation of evidence operation that is thought to underlie the behavioral performance in this task (Pinto et al. 2018). Our models further identify two different network mechanisms by which evidence can be accumulated by, and transferred between, sequentially activated neural populations, each of which is associated with a different shape of neuronal tuning curve (monotonic vs unimodal). The data analyses then demonstrate a dichotomy in the tuning between hippocampus and cortical regions, similar to that observed in our models with, interestingly, retrosplenial cortex showing some of each type of coding, consistent with its strong anatomical connections both to other neocortical regions and to hippocampus.

We note that a major caveat in all of this work is that, in the absence of perturbation experiments, we cannot conclude if the recorded regions are an active part of the accumulator or a passive downstream readout of the accumulator (and we note that the anatomical location of the integration in accumulation-of-evidence based decision-making tasks remains one of the longest-standing, open questions in the field of decision-making). Furthermore, a readout of the neurons performing the evidence accumulation may transform a graded representation of analog evidence into a more choice-like readout. For these reasons, our primary focus in comparing the models to the data was (and still is) to distinguish the monotonic from unimodal tuning curves that correspond to the two different classes of models.

This all being said, we completely agree with the reviewer that it is interesting and useful to delve further into the data to more precisely characterize the recorded activity, and to better distinguish, when possible, if the recorded neuronal activity carries a graded representation of evidence or only a choice-encoding representation such as could occur through a readout of the actual accumulator neurons. We also agree that it is useful to include choice-encoding populations in our models, to better connect to the data. We therefore, as described below, have expanded our data analyses to better characterize the representations of evidence in the recorded brain regions, including at different positions in the maze, and expanded our models to include some possible means by which choice-encoding neurons, and their relative numerosities at different positions in the maze, could be generated.

In the areas showing ‘monotonic tuning’ (RSC and ACC) neurons seem to largely represent the best current categorical guess (Figs. 5 and 6). Although the authors argue that linear evidence accumulation transitions to binary choice encoding (lines 532-533) their Fig. 6D,F shows that, at best, the small fraction of neurons that have non-zero regression coefficients for either variable is similar. Then only binary choice becomes significantly represented later in the corridor. Furthermore, even at these early positions we are talking about ~ 2% of neurons, and also, at these early positions distinguishing between binary choice and graded evidence becomes more difficult (in the limiting case there only one tower has been seen they cannot be distinguished).

In exploring representations in ACC and RSC, our main finding was that these regions had monotonic tuning to evidence. However, as we noted in the paper and the reviewer correctly emphasizes here, there are multiple possible forms of monotonic tuning with different functional interpretations and the original analysis only found a small percentage of neurons that could be significantly identified as graded beyond binary sign-of-evidence.

These helpful comments led to our having multiple realizations about the limits of the previous single-neuron encoding analyses and why they may have provided an underestimate of the amount of graded-evidence encoding in the population as a whole. First, most fundamentally, we realized that our models do not require that individual cells show graded responses, but only that the population as a whole represents accumulated evidence, since the units of our models should be thought to represent populations rather than individual cells. Such graded representations could result from either a collection of graded evidence cells (as in the present model) or a collection of cells which step at different firing rate thresholds (which would be a trivial variation on the present model). Thus, the exact dynamics of individual cells is not a critical feature of our model, only that populations have graded representations of evidence. We have clarified this point in the text on p. 17. Second, we realized that our previous single-neuron encoding analysis neglected one of the most conventional, and powerful, ways of distinguishing between neurons that encode for behavioral choice, as opposed to accumulated evidence: the use of incorrect trials to determine which neurons follow evidence versus which neurons flip their tuning (firing rate as a function of evidence) on incorrect trials.

Building on these observations, to better probe for graded evidence representations in the data, we take the following set of approaches. First, we discuss further limitations of our original single-neuron encoding analysis and present the modified single-neuron encoding model that uses incorrect trials to better account for neuronal firing that follows behavioral choice, as discussed above. Second, we introduce two new population-level analyses that further expand on these findings: single-trial decoding of evidence when controlling for the current best choice and cue encoding models to look for cue responses consistent with integration.

(i) Single-Neuron Evidence vs. Choice Encoding Analysis

In the original manuscript, we only analyzed correct trials and then attempted to differentiate cells that respond in a step-like manner to the sign-of-evidence, i.e. cells

representing the “best current categorical guess,” from more linear representations of graded evidence. Our original analysis demonstrated that a small percentage of neurons significantly carried graded information beyond the sign of evidence. Such a small fraction could reflect that evidence is encoded more in the number of neurons firing across the population (rather than individual neurons’ graded firing rates), that only a small fraction of neurons are needed to perform evidence accumulation, or that the recorded region is a downstream readout of an upstream graded representation (and, as noted above, pinpointing the site of accumulation continues to be a challenge in the field). Alternatively, this result could be due to experimental factors, e.g. that our single-neuron data were too noisy or the Ca^{++} indicator was too nonlinear to easily distinguish evidence and choice representations, therefore leading to an undercounting of the number of neurons carrying evidence information beyond choice (more on this latter possibility is provided below).

Moreover, we agree with the reviewer that the definition of these two response patterns is the same for cases where there is only one tower (and in our original results, we note that neither coefficient showed significance up to the point where more than one tower was introduced). Importantly, this could lead to undercounting of neurons that encode evidence, because the two variables are equivalent early in the maze.

To address this limitation, we took a complementary approach to identify neurons that respond to evidence beyond choice. Our revised model fits single-neuron responses on both correct and incorrect trials to evidence and behavioral choice. This revealed a substantial population of cells that correlates with evidence (beyond choice) during the cue period, followed by a transition to choice encoding neurons dominating in the delay period. This result is consistent with our simpler observation in the original manuscript that there are more neurons with graded neuronal tuning curves early in the maze (after the very earliest period where there is only 0 or +/-1 unit of evidence, so that gradation cannot be determined) and more with step-like tuning curves later in the maze (Fig. 5 A,B,I,J, Extended Data Figs. 6, 7, 9).

Fig. 6 F, G, H (F) Evidence vs. choice single-neuron encoding analysis. Top: regression equation. Bottom: schematics of the firing across positions and evidence levels for a purely evidence encoding cell and a purely choice encoding cell on correct trials. (G) For each position, the average fraction of active neurons in each session in ACC (defined as neurons with fit position mean within one position standard deviation of the animal’s location) with significant coefficients for evidence (crimson) or choice (lime) in the choice vs. evidence encoding model, based on the F-statistic. Error bars indicate s.e.m. across sessions. The horizontal dashed gray line indicates chance level (1%). (H) Same as (G) but for RSC.

Given that this updated model uses a more relaxed form of evidence dependence than the previous model, we tested whether the responses in the population identified as encoding evidence beyond choice were graded. To do so, we plotted the evolution of the population firing rate on single trials separated by final evidence. We observe that this subpopulation shows graded firing with final evidence during the cue region (Extended Data Figure 13E,F).

Extended Data Figure 13 E,F. (E) Trial-averages of the difference of mean activity in the left and right populations of active cells in ACC for trials with different final evidence levels (indicated by the color bar). (F) Same as E but for RSC.

With regards to the use of Ca⁺⁺ imaging as a proxy for neural activity, we believe calcium imaging is well suited to differentiating between monotonic and non-monotonic tuning and to identifying choice-encoding cells that change their direction of response on incorrect trials. However, the nonlinearities present in calcium imaging may make differentiating between different forms of single-neuron monotonic tuning more challenging. In fact, in an electrophysiology dataset (manuscript in preparation) recording from ACC in the same task, we see a much more robust graded evidence representations (Fig. R2.1).

Figure R2.1 Examples of linear evidence encoding neurons in electrophysiology data. (A) Left: Example average firing across trials binned by final evidence over the course of the maze for an example ACC neuron. **Right:** Evidence tuning curve of the example cell for correct (black) and incorrect (dashed gray) trials. **(B) Tuning curves of the subpopulation of cells in ACC identified by an evidence vs. choice encoding model as ipsi- (pink) or contra-prefering (teal) for correct (left) and incorrect (right) trials. (C) Additional example neurons identified as evidence encoding by the evidence vs. choice encoding model.** All recordings were performed with neuropixels probes during the accumulating towers task. Ipsi- and contra- indicate whether cues appeared on the same or opposite side of the recording site.

(ii) Population Evidence Decoding

To further demonstrate the presence of graded evidence coding in the population, we now include new analyses to demonstrate that evidence can be linearly decoded from the population beyond chance (as defined by a current-best-choice matched null distribution). By only testing decoders within the same current-best-choice trials, we eliminated the possibility that correlations between true and decoded evidence in the middle and late portions of the cue period resulted merely from correctly identifying the sign-of-evidence (see also response to Reviewer 1). While decoding performance drops at later positions in the cue period, it remains above chance, suggesting graded coding beyond binary choice. We have added the results of this analysis to Figure 6 (panels J,K reproduced below). We also note that such decreases in the ability to decode evidence over the course of the maze are consistent with recent findings in an auditory evidence accumulation task that showed that neural dynamics change from an evidence-accumulation regime to an “unresponsive to new evidence” regime at the inferred time of commitment to a choice (Luo et al. 2023).

Fig. 6 J, K (J) At each position, the cross-validated correlation between actual evidence and predicted evidence from a linear population decoder in ACC on correct right evidence ($e \geq 0$) trials (red) and left evidence ($e \leq 0$) trials (blue), compared to a shuffle control (grey). Error bars indicate s.e.m. across sessions. **(K)** Same as J but for RSC.

The reviewer correctly points out that early in the cue period where we see the largest peak in our decoding, observed evidence values will be 0 or +/-1 so that graded and binary evidence representations are indistinguishable. For this reason, we also include a supplementary analysis where we remove timepoints with evidence 0 from the dataset (Extended Data Fig. 13 G,H). Even though this reduces the size of both our training and test sets, and thus makes our results noisier, we find that the decoder continues to perform above chance during the cue period, falling to chance levels in the delay period.

Extended Data Fig. 13 G,H (G) At each position, the cross-validated correlation between actual evidence and predicted evidence from a linear population decoder in ACC on correct right evidence ($e > 0$) trials (red) and left evidence ($e < 0$) trials (blue), compared to a shuffle control (grey). Error bars indicate s.e.m. across sessions. **(H)** Same as G but for RSC

(iii) Population Cue Encoding Model

To complement this decoding analysis, we also fit a cue encoding model to test whether the neural population showed prolonged cue responses that are necessary for integration. In this model, cues are convolved with a response kernel to predict the difference of right-preferring and left-preferring population activity (Fig. 6 A-E).

Fig. 6 A-E (A) Schematic of the cue encoding model, where each left and right cue is convolved with a left or right cue response kernel, K_L or K_R respectively, to predict the population activity. (B) Example single-trial population activity (black) compared to the model predicted activity (dashed gray) in ACC. (C) Same as B but for RSC. (D) Cue kernels fit from the cue encoding model to right cues (red) and left cues (blue). (E) Same as D but for RSC.

Overall, while our analyses rigorously demonstrate that graded evidence coding is present in the population, we agree with the reviewer's interpretation that choice tuning seems to account for a large proportion of the neuronal activity, especially later in the maze. To better capture this fuller picture of the data with our models, we have expanded our models to show possible ways that choice tuning could emerge late in the maze (see Figure 7 below and response to later comment for full details).

In HPC, convincing evidence is shown that a significant proportion of neurons have 'bump shaped' tuning curves for evidence, but this is also fairly indirect evidence that the population as a whole is tracking the running accumulated evidence in the task. There are certainly more direct ways to probe if the HPC is representing instantaneous accumulated evidence or not.

We thank the reviewer for their question that prompted us to examine decoding of evidence from the HPC data. Previous published analysis (Nieh et al. 2021) of this dataset has demonstrated that evidence can be decoded from a nonlinear manifold (calculated from MIND (Low et al. 2018)), constructed from population activity. We independently verify this claim using a different nonlinear manifold technique (CEBRA (Schneider et al. 2023)). We show that, from embeddings learned on a training set, we are able to decode evidence from an independent test set above a choice-matched shuffle (Extended Data Figure 11). We have modified the manuscript to include a discussion of decoding evidence from the population in HPC and a supplementary figure with these results (Extended Data Figure 11). This figure also demonstrates that, unlike in the neocortical regions, evidence in the hippocampus continues to be decodable beyond chance into the delay region.

Extended Data Figure 11. Decoding evidence from HPC populations. (A) CEBRA embeddings uncover a nonlinear mapping that smoothly captures evidence levels (indicated by the color of the points) in the training set. (B) Same as in (A) but for the test set. (C) A k-nearest neighbors decoder can decode evidence from the CEBRA embedding (average performance measured by Pearson’s correlation coefficient (r) between the actual and decoded evidence, blue) above a sign-of-evidence matched shuffle (gray). Lines indicate performance on individual sessions, compared to the corresponding shuffle. (D) Average decoding performance across different positions in the maze compared to a sign-of-evidence matched shuffle (gray) for 2-dimensional CEBRA embeddings.

We note that the correlation between actual evidence and predictions on the test set remain relatively low (similar to the findings in Nieh et al. 2021). This low performance could result from the fact that the hippocampal cells narrowly respond to particular evidence levels; thus, if we do not record from a cell corresponding to evidence level e , then it will be difficult to distinguish e from other non-recorded evidence levels. Furthermore, previous analysis of this dataset found that, in addition to encoding evidence and position, the hippocampal code is tuned to at least one additional, apparently non-task-related dimension (Nieh et al. 2021). This makes the neuronal responses even sparser since neurons only respond when they are also in the response field of this non-task-related dimension, and thus makes accurate evidence decoding even more challenging.

Thus, the bulk of the constraints that are used to develop the models derive from functional considerations that are not experimentally demonstrated. If, for instance, the data in Figs 1,5-6 was presented first, it would be more salient that the modelling in Figs 3,4 only loosely maps to

the data. What the data convincingly demonstrates is a dichotomy in terms of the shape of tuning curves to evidence, but this fact, without the requirements for graded evidence accumulation, provides a much weaker constraint on the mathematical models. If the authors wish to make a tight link between data and modelling, less constrained models (presumably a larger model class) should be considered. If the authors wish to explore possibilities for modelling sequential graded accumulation of evidence with a mostly theoretical motivation, then the theoretical and experimental aspects of the manuscript become less tightly associated. It would need to be made explicit that the data relates to these models fairly indirectly, and instead bears on a much larger model class, which seems to partly defeat the purpose of having the data and models together as it is done in the present version.

As discussed in the overview to the Reviewer 2 comments (p. 17), we agree that the main point was to construct two models of the accumulation of evidence with sequences and that the primary result from our data analysis is the finding of the dichotomy of tuning curve shapes that could result from different sequence-based accumulation mechanisms or a readout of them. We also hope that the data analyses above make clearer the experimental support for the encoding of graded evidence in a significant portion of the recorded population. Importantly, it was not known *a priori* which if any of the analyzed regions represent accumulated evidence, so we thought it was interesting that ACC largely has a monotonic representation, HPC largely has a unimodal representation, and RSC shows a mainly monotonic representation (though with some unimodal cells). We also, due to the issue discussed above of not being able to distinguish an accumulator region (or set of neurons within a region) from a readout of the accumulator, found it useful to make predictions for perturbation experiments that, once a candidate accumulator is identified, could probe the evidence accumulation process itself. That being said, we completely agree with the reviewer that there are many cells in each region not accounted for by the models, most notably the large population that encodes binary choice.

We think the presence of these choice cells and the transition to an increasingly binary representation across the maze are the two major features of the data that were not captured in the original paper's models. To address this gap, we have added a new Figure 7 that presents models of two (not mutually exclusive) mechanisms that would reproduce this shift from evidence to choice representations. First, there could be a population of choice cells at each position that reads out choice from the evidence cells at each position, with larger numbers of these choice cells at later positions in the maze (Fig. 7A-C). The larger number of cells tuned towards the end of the maze results in an increased amplitude in addition to sharpening of the population tuning curve later in the maze. Second, if the accumulator itself is slightly unstable, the evidence tuning will become more step-like later in the maze (Fig. 7D-F). We believe the inclusion of these models tightens the link between the model and the data.

Figure 7. Simulated shifts towards choice tuning in the population. (A) Model schematic of a population of choice cells that readout from the accumulator and indicate if there is more left evidence (navy blue cells) or right evidence (dark magenta cells). The number of neurons in the choice readout population tuned to each position grows along the chain. **(B)** Tuning curves of the choice cells at their preferred position. **(C)** As in Figure 6, population average tuning curves of the entire population of

evidence-tuned cells including both cells in the accumulator and the choice-readout cells. Cross sections show a shift towards choice tuning with increasing position in the maze. **(D)** Schematic of the mutually inhibiting chains model with unstable tuning, in which the strength of self-excitation increases with position along the chain. **(E-F)** Same as B-C but for cells in the unstable mutually inhibiting chains model.

2. Schematic and somewhat unrealistic modelling approach. While I appreciate the effort of translating the desiderata into specific mathematical implementations, the models presented seem somewhat artificial. A large number of tuning conditions need to be satisfied. The treatment of the all-or-none position signal is also quite artificial. The data in Fig1 shows graded and heterogeneous spatial ‘tuning curves’ and connected neurons will interact throughout the corridor, not just through the ‘position gate’.

We thank the reviewer for their attention to the fine-tuning conditions needed for perfect integration. These fine tuning parameters come in two classes, corresponding to the two fundamental computations performed by our models: (i) tuning conditions needed for accurate accumulation of evidence within a position and (ii) tuning conditions needed for faithful transfer of accumulated evidence between positions. In our response, we consider these two classes separately.

(i) Tuning Conditions for Accurate Accumulation of Evidence Within a Position

As we noted in the Competing Chains Models section, “As in the case of traditional, non-sequential accumulator models, perfect accumulation up to this upper bound requires fine-tuning of the parameters to prevent leak or uncontrollable growth in the integrator.” Such conditions have been well studied in these models (for review, see Goldman, Compte, & Wang 2009), and we have rederived the conditions for perfect integration in the Supplementary Text. As seen in the tuning curves from the mutually inhibiting chains model (Fig. 3E), this integration need not be perfect across the entire range of observed evidence levels to perform the task perfectly. Moreover, in the case of the competing chains models, integrated information just needs to be maintained such that the chain corresponding to the side with more evidence has greater amplitude, and in the case of the position-gated bump attractor, the bump needs to be maintained in the side of the population corresponding to the side with more evidence.

Much work has explored how to increase robustness in each of these styles of models. Such fine tuning conditions can also be reduced by introducing corrective feedback mechanisms into the circuit (Lim & Goldman 2013, Lim & Goldman 2014) or by introducing bistable processes into the population of neurons or single-neuron biophysics (Koulakov et al. 2002, Goldman 2003).

We have added the following text to better emphasize this point:

“Previous studies of such networks have also suggested that such accumulator networks may be made more robust to these fine-tuning conditions through incorporation of bistable

components (Koulakov et al. 2002, Goldman 2003) or corrective feedback mechanisms (Lim & Goldman 2013, Lim & Goldman 2014).”

In the case of bump attractor models, robustness to noise to prevent diffusion and drift has also been extensively studied (Camperi & Wang 1998; Wu, Hamaguchi, & Amari 2008, Burak & Fiete 2012) and may be increased in robustness through error-correcting codes (Sreenivasan & Fiete 2011) or extending representations to multiple bumps (Wang & Kang 2022). Other work has also explored ways to allow for parametric heterogeneity in these models (Renart, Song, & Wang 2003, Darshan & Rivkind 2022), which may particularly increase robustness of discrete states (Kilpatrick, Ermentrout, & Doiron 2013) as in the case of discrete evidence levels.

We have added text to provide these additional citations for readers especially interested in the robustness of our proposed models:

“We also note that our assumption of perfect symmetry simplifies the analysis, but is not required to produce bump attractor models (Renart, Song, & Wang 2003, Darshan & Rivkind 2022) and that the robustness of bump attractor models to diffusion and drift of the bump in the presence of noise has been extensively studied (Renart, Song, & Wang 2003, Camperi & Wang 1998, Wu, Hamaguchi, & Amari 2008, Burak & Fiete 2012) and may be reduced through error-correcting codes (Sreenivasan & Fiete 2011), extending representations to multiple bumps (Wang & Kang 2022), or relaxing the continuous attractor assumption to have only a discrete set of memory states (Kilpatrick, Ermentrout, & Doiron 2013).”

(ii) Tuning Conditions for Faithful Transfer of Accumulated Evidence Between Positions

Our models introduce another set of tuning conditions for the transfer of information between positions.

For the competing chains models, as outlined in the Supplementary Text, we require that the rate of decay of information at the previous position is the same as the rate of accumulation at the next position to prevent information loss. These fine tuning conditions could be relaxed if we do not require that firing rates be preserved between positions but only the relative information.

The conditions on the position gating are much less sensitive for both classes of models, merely requiring that the gating signal bring the neuron above threshold and that neurons tuned to the same position receive the same signal. It is the fact that this gating signal only brings active neurons above threshold that prevents neurons from “interact[ing] throughout the corridor,” i.e., although neurons are synaptically connected the inputs conveyed between these neurons remain below threshold. For simplicity in our simulations, we used a step-like position gate of the same width for each position. Neither the step shape nor the homogeneity of the gating signal are necessary for our model, and we have added simulations showing that smoother gating functions such as a Gaussian (Extended Data Fig. 2G) and gating with heterogeneous widths (Extended Data Fig. 2H) both still accumulate evidence and produce

choice selective sequences. Parallel chains with such heterogeneous tuning (as previously suggested to occur for faster timescale sequences (Okubo et al. 2015)) would give rise to the “overlapping tuning curves” the reviewer points out in the sequence plots (which pool neurons from multiple animals). Alternatively, a more steplike gating signal could result from a winner-take-all decoder of a bump model that integrates velocity (Extended Data Fig. 2I) and previous work has explored heterogeneous tuning in such bump networks that could lead to heterogeneous widths of the position readout (Darshan and Rivkind 2022).

Extended Data Figure 2G-I. Uncoupled competing chains model of evidence accumulation through sequences. (G) Choice-selective sequences (as detailed in Extended Data Fig. 2C) for the mutually inhibiting chains model with a Gaussian position gating signal. **(H)** As in G, but for the mutually inhibiting chains model with square position gates of different widths. **(I)** Schematic of a velocity integration method for generating square pulses, where velocity is integrated through a bump attractor to produce cells representing different preferred positions. A winner-take-all network takes these position cells as input to produce a square position gate.

Some specific issues I had with the modelling:

2.1 I might be mistaken, but it seems to me like the formula between lines 113 and 114 in the Supp. Text is incorrect? The upper limit of the integral is P_0 , but it should be quantity with dimensions of time. Presumably P_0/speed , i.e., the time during which the position gate is open? Given the values of these parameters, $a \cdot P_0/\text{speed} = 1 \text{ Hz} \cdot 20 \text{ cm} / 50 \text{ cm/s} = 0.4$, so that each position is actually integrated quite partially. As the authors point out, full integration would require this quantity to be $\gg 1$.

We thank the reviewer for their careful read of the supplementary text. The reviewer is correct that the upper limit of the integral should be in units of time. We have changed the upper

limit (as correctly identified by the reviewer) to P_0/v for an animal traveling at speed v . Then, the decay rate is given by $a \cdot P_0/v$. In the original manuscript, we mistakenly reported a without correctly converting units from cm^{-1} to s^{-1} ; the correct value is $a = 1 \text{ cm}^{-1} = 50 \text{ s}^{-1}$ (assuming a constant velocity of 50 cm/s). Thus, the decay rate in our simulations is 20, $\gg 1$ as required.

2.2 Why does the neural activity profiles in figs 3C, 4C look so discontinuous along space? If the decay rate of activity is 1Hz and the mice runs at 50cm/sec (see comment below on speed), then graded decay should be evident on a 300 cm region.

We thank the reviewer for this question, which points to the fact that we did not correctly convert the units of our simulations from position to time when reporting the parameters of our models. Since we assumed a constant velocity in our simulations, we performed our simulations as a function of position rather than time. When reporting the time-based differential equations, we reported the parameters used in the simulation without correctly converting the units from centimeters to seconds. Thus, the decay rate of activity should be 50 Hz, and would be evident over 1 cm but not 300 cm. We have now corrected the parameter values reported in “Parameterization of the Mutually Inhibiting Chains Model” and “Parameterization of the Position-Gated Bump Attractor” in the main text as well as “Parameterization of the Uncoupled Competing Chains Model” in the supplementary text.

2.3 What is the role of speed on the activity of the network? If the position gate is fixed integration of evidence will proceed more slowly for faster mice? Is this reasonable?

We thank the reviewer for this question about the role of the animal’s speed on integration in the models. The integration in our models is robust to the speed of the animal up to the point of an animal running “too fast”. As noted in the Supplementary Text (lines 112-118 and lines 202-206), advancing too quickly could result in a loss of information between positions in the chains based models. This does not mean that the evidence accumulation proceeds more slowly, but rather that the integration would be “leaky” for a mouse traversing the maze more quickly (less information from neuron $i-1$ is integrated by neuron i before the animal advances to position $i+1$). Previous analysis shows that there is little correlation between speed and performance within a given session, while average performance may in fact be better for mice that run on average faster (see Supplementary Fig. 7 in Pinto et al. 2018), suggesting that the mouse speed is appropriately tuned to the duration of the position gate.

Alternatively, by introducing additional connections such that information is propagated to the active neuron from throughout the chain, integration can occur without information loss, independent of speed. For readers interested in running speed, we have added to the Supplementary Text a reference to Pinto et al. (2018), as well as a sentence regarding the potential role of having additional connections:

“Previous analysis of mouse running speed versus performance has shown little correlation within a session and a positive correlation on average across all sessions (see Supplementary Figure 7 in Pinto et al. (2018)), suggesting that mice do not run at such a fast speed that

information loss is problematic. Alternatively, additional connections could support the transfer of information from longer distances in the chain, increasing robustness to information loss.”

2.4 What is the shape of the dynamics of the sum mode in the competing chains model? Are there any qualitative features that could be compared with the data? The global activity of the network (and/or evidence-tuned neurons is readily accessible).

For a network to integrate along the difference mode, activity along the common mode should decay, suggesting that such a signal should only be transiently detectable in the data. At any given position in the chain, the common (sum) mode input is the position gating signal, $P_i(t)$. Like in traditional competing accumulators, the eigenvalue ($b-a-e$) associated with this mode is negative (due to the conditions satisfied for the difference mode to have eigenvalue 0), such that activity along this mode will decay exponentially. We have now made this explicit in the Supplementary Text:

“For a perfect integrator that accumulates evidence in the difference of firing between the two chains, we require that the eigenvalue corresponding to the difference mode eigenvector is zero, giving $e = a - b$. Due to the mutual inhibition architecture, the eigenvalue corresponding to the common mode ($b-a-e$) is negative (corresponding to decay).”

Our mathematical analysis shows that the common mode input is needed to raise neural activity above threshold for integration, so that the disinhibitory positive feedback loops corresponding to the difference mode can operate. Consistent with this prediction, neurons show a gradual rise to their maximal position tuning, for both their preferred and non-preferred choices, as seen in the sequence plots (Fig. 1B-E) and the two dimensional tuning curves of individual neurons (Fig. 5A,E,I, Extended Data Figs. 7-10). More direct tests of the dynamics of this mode would likely be inconclusive for a number of reasons. One reason is that the shape of the common input is unknown - we show in our analysis that the shape of this common input need only be above threshold and the same for each chain. Another reason is that we expect that the decay dynamics of this mode should be fast and there could be many other common processes contributing to dynamics on the same timescale.

Overall, readers will benefit from a more thorough characterisation of the models. More generally, while the models seem like a good first step, they seem too schematic and unrealistic for a finished product, specially given the fact that they are structurally unstable and the adding both functional heterogeneity (present in the data) and parameter heterogeneity (present in the brain) is not simply a matter of cosmetics, but rather represents an important challenge to the desired model function.

As discussed in responses to the previous comments, the fine tuning conditions present in our models are not unique to this paper but have been extensively studied for both competing accumulators and bump attractor networks with approaches to allow for more realistic heterogeneity and robustness to noise. As made explicit above, we now point to these other works in our discussion of our models.

Moreover, the units in our models should not be thought of as single neurons but instead representative of the average responses of populations of similarly tuned neurons, such that individual cells may still exhibit parameter heterogeneity. When modeling large numbers of neurons, such heterogeneity tends to have very little impact on the overall dynamics because the leading eigenvalue controlling the integrating dynamics is robust to such heterogeneity (Cannon et al. 1983, Seung 1996).

Other comments:

1. Fig 2 seems expected. Maybe it could go to the supplementary info.

We agree with your point that these results are expected for someone with intuition about these models. However, we have found in presenting this work to a general audience that this figure is useful in demonstrating the gaps in existing models, as well as introducing the two canonical models that are the building blocks of our sequentially based models. For this reason, we would prefer to keep the figure in the main text, but we could easily move it to the supplement if you and the editor prefer.

2. The optogenetic perturbation simulations in Figs 3 and 4, while interesting, take a lot of space and are not pursued experimentally. The authors might consider again moving them to the supplementary information and using that space to describe more precisely how the models work (in a sense deconstructing panel C by showing temporal profiles of synaptic currents from different sources etc), effect of different parameters, etc

We thank the reviewer for this suggestion. We have now added a panel to illustrate the temporal profiles of different sources of input to allow the reader to build more intuition regarding the terms of the model. We have modified both Figure 3 and Figure 4 to include these panels.

New Panels for Fig. 3 (left) and Fig. 4 (right). (Left, Fig. 3B). Top: Firing rates of neurons in the left (blue) and right (red) chains of an example trial for positions 40 to 80 cm. Vertical dashed lines indicate the location of left (orange) and right (purple) towers. Background shading shows the transition between different active positions. Middle: Position gating signal at each position. Bottom: Input current to the left (orange) and right (purple) chains resulting from the external cues. **(Right, Fig. 4B)** Top: Each row shows the firing rate of a subset of neurons from 40 to 80 cm in an example trial of the maze. Neurons are first

sorted by position tuning (black horizontal lines denote divisions between positions) and then by evidence tuning within a position. Dashed vertical lines indicate the position of the left (orange) and right (purple) cues. Background shading indicates the active position. Bottom: Position gating signal at each position.

Due to the value of future optogenetic experiments for testing the predictions of the models and differentiating between the variants of the competing chains models (see response to Reviewer 1, minor comment 1), we would prefer to keep these panels in the main figure. We believe that these panels also help build intuition regarding the synaptic connections between cells and how changes in the firing rate of one cell impacts the others, as asked about in this comment.

3. In lines 571-2 the authors speculate that the position signal (which in the model is totally external), might come place cells in the hippocampus, but the authors are actually modelling the hippocampus. This seems strange.

We thank the reviewer for this question about the origin of the position signal. We provided hippocampal place cells as one possible hypothesis, given that the existence of these cells is well-established. Of course, there are many possibilities. We have expanded the paper to consider two additional possibilities:

First, there are many cells in each region (including the hippocampus) that show position but not evidence (or choice) selectivity, which could locally provide this position gating signal. Since our primary focus in this work is on evidence accumulation, we did not initially show these cells (previously shown for RSC in Koay et al. 2022 and HPC in Nieh et al. 2021) but have now added these cells to Extended Data Fig. 1. Such cells could be a readout of a path integration process.

Extended Data Figure 1 with added non-preferring cells.

Second, the hippocampus could simultaneously accumulate evidence and position, so we now include a modified sequence-based bump attractor model, where evidence is accumulated along the evidence axis as in our original model and velocity is accumulated along the other axis to give rise to a joint position-evidence representation (Extended Data Fig. 3).

Extended Data Figure 3. Planar bump attractor for evidence accumulation through sequences. (A) Schematic of the neural circuit architecture for the planar bump attractor. Each row represents neurons (circles) that respond at a given position. Within a row, each neuron represents a different evidence level, ranging from left-most to right-most. Blue: left-preferring neurons; red: right-preferring neurons. Like in the position-gated bump attractor, the bottom row of neurons illustrates the cue-related inputs and connectivity within any given row: local excitatory (green) and broader inhibitory (brown) connections between neurons as well as external inputs to the circuit from the left (orange lines) and right (purple lines) towers via the corresponding shifter neurons (orange and purple circles). The rightmost portion of the diagram illustrates the velocity-related inputs and connectivity within any given column: local excitatory (green) and broader inhibitory (brown) connections between neurons at different positions as well as forward (pink) or backward (cyan) external velocity inputs that enter the network through corresponding position shifter neurons (pink and cyan circles). **(B)** Each row shows the non-normalized firing rate of a model neuron at each position in the maze, averaged across simulated trials without input noise when the greater final amplitude was in the left (left choice, left column) or right (right choice, right

column) chain. Neurons were divided based on their choice-selectivity (see Methods) and ordered based on the position of peak activity. **(C)** Tuning curves of a subset of individual neurons to evidence, calculated when the neurons have peak position activity, for left-preferring (blue) and right-preferring (red) neurons. **(D)** Example simulation of the planar bump attractor for a theoretical trial where the animal can move both forward and backward. Top: Raster of activity of individual model neurons across different positions in the trial. Incoming towers are indicated by the dashed orange (left cue) and purple (right cue) lines. Bottom: The velocity of the animal at different points of the trial, including forward (pink) and backward (cyan) movement.

We thank the reviewers for their positive feedback and recommendation that the manuscript is now “suitable for publication”. All reviewers found that we “responded in detail,” (Reviewer 1) “performed a substantial amount of work” (Reviewer 1), “largely satisfactorily addressed [their] concerns” (Reviewer 2), and “thoroughly and carefully answered all outstanding questions” (Reviewer 3), with Reviewer 3 even praising that our responses “have gone above and beyond.”

The reviewers’ comments (black italics) and our individual responses (blue) to each point are included below.

REVIEWER COMMENTS

Reviewer #1

The reviewers responded in detail to my concerns and have performed a substantial amount of work in the revision. In my opinion, some of the responses, such as the addition of a comparison of correct and incorrect trials were useful and added to the paper. Others, like the clarification of regarding the use of “evidence-tuned” were important clarifications.

Thank you for your positive feedback on these key revisions to the paper.

However, my main concern, adding more single trial analyses, was less well answered. My comment was that there should be aspects of single-trial dynamics that are quite different between the two types of models. Just as the authors describe perturbation experiments that would better differentiate between the models, there should be single trial analyses that do the same. For instance, the authors write that (starting line 384): “This is because the random walk of the bump location as evidence is accumulated leads to a greater range of evidence levels being observed at later positions in the trial; thus, on average across trials, there are more different neurons active at later positions, with each neuron corresponding to a different possible evidence level”. This an example property that could have been better probed with a single trial analysis. I don’t believe this is the best example, I just mention it as an example of some single-trial analysis that makes expected model properties (sharper bumps) more apparent.

With regards to the prediction on line 384, this was a description of our models rather than the data. For the data set we are considering, we could not perform this simple analysis because, as noted in our previous reply, confounds exist relative to traditional, non-sequential models. In particular, these analyses are more challenging because: (1) the set of neurons firing at any given position changes as the animal progresses down the maze, (2) the number of cells recorded at a given position can change between positions, and (3) our data are sparse so that there are very few neurons at many position-evidence levels. Together, these data sampling issues mean that it is difficult to tell whether there are ‘more different neurons active at later positions’ on a single trial, as can be seen in the models.

My suggestion was for one or more analyses along such lines that clarify and then contrast properties of the two suggested models. The population analyses performed instead were more focused on population-level versions of overall properties. Analysis (i) shows that evidence can be decoded (which is not surprising given the single neuron results) analysis (ii) looks for sustained responses, which unless I am mistaken have to exist in both models and analysis (iii) is more of a visualization that better reveals graded encoding. These are all interesting, but do not address my specific comment which is on single trial dynamics that differentiate the two models. I appreciate the authors’ response that drift is tricky and I fully appreciate that analysis and some others can’t be done given the recordings available. My comment referenced that as

one specific example of a broader set of analyses which I believe would have strengthened the paper.

Thank you for this comment. First, with regards to the use of population level analyses for single-trial analysis, we are not sure if this is in contrast to performing single-neuron analyses but we were responding to the previous reviewer comment that expressed being “*surprised that there was almost no single-trial population level analysis*”. In any case, we used population-level analyses because, especially for looking at single trials, the analysis of the population as a whole has more statistical power than looking at single neurons individually.

We also apologize for any confusion on our part regarding the goal of your question. We had thought the differences in single-cell encoding as they relate to the differences between the two models were clear from the single-cell tuning curves. Since you had alluded to the nature of the drift, we had assumed the question was meant to probe differences between monotonic models driven by graded evidence and choice (ramping vs. stepping) rather than the difference between the competing chains and position-gated bump attractor classes. That said, analysis (ii) (the cue encoding model) does differentiate the competing chains models and the position-gated bump attractor. In the position-gated bump attractor, we only expect a change in the magnitude of the population firing for cues that change the sign of the evidence, while other cues lead to no response. As a result, the HPC data is not well fit by this model ($R^2 < .05$). Consistent with this, we found that general cue responses were many times smaller in HPC than in ACC or RSC. When we then broke down the responses by current evidence level, the HPC responses were only significantly different from zero (although still smaller than in ACC or RSC) for cues at low evidence levels, where a cue can change the sign of evidence. Thus, analysis (ii) is a single-trial analysis that distinguishes the monotonic and non-monotonic models. However, given that the cue-encoding model doesn't fit the HPC data well, and that the tuning curves are so starkly different for HPC (non-monotonic) versus ACC (monotonic) and RSC (primarily monotonic), we thought that it was more confusing than helpful to show analysis (ii) applied to the HPC data.

That being said, I believe the paper is suitable for publication.

We're glad that you recommend the paper for publication.

Two more minor comments: 1. I agree with reviewer 2 that Figure 2 is not particularly useful and can be sent to supplementary.

Thank you for your input. While we agree with the reviewers that the results in Fig. 2B are clearly expected for readers familiar with integrating models, we found Fig. 2D less immediately intuitive. We have found this figure to be particularly helpful in presenting this work to less expert audiences. Moreover, given the lingering confusion regarding the model schematics, we have elected to leave this figure in the main text since it shows the connectivity at each position of the sequence-based models. We also now refer to Fig. 2A and 2C when discussing the models to help readers digest the large number of connections.

2. I personally find the model schematics very difficult to parse. To me the equations and verbal descriptions were more useful. While I imagine the authors have already dedicated some thought to this, it would be nice if they could find some better schematics (which can be for instance tested by asking people not directly involved with the study what they infer from the schematics).

Thank you for this comment. We agree that there are a large number of connections present in these panels. In addition to referencing Fig. 2A and 2C to help clarify the connectivity, we have also reduced the number of units shown in these figures and rotated the schematic in Fig. 3A so that it is consistent with the other schematics in having left-encoding neurons on the left of the figure and right-encoding neurons on the right.

Reviewer #2

The authors have largely satisfactorily addressed my concerns.

Thank you. We're glad that you found our responses addressed your comments.

Reviewer #3

I have been brought in as an additional reviewer following missing responses from R2 in this review process. After reading the responses provided to R2's concerns, the authors have thoroughly and carefully answered all outstanding questions. R2's general critique was slightly vague at times, yet the authors have gone above and beyond, congratulations.

Thank you for stepping in as an additional reviewer, and for your positive assessment of our responses to R2's comments.